# A cryogenic on-chip microwave pulse generator for large-scale superconducting quantum computing

Zenghui Bao[1,3], Yan Li[1,3], Zhiling Wang[1], Jiahui Wang[1], Jize Yang[1], Haonan Xiong [1], Yipu Song[1,2], Yukai Wu [1,2], Hongyi Zhang [1,2] & Luming Duan [1,2]

For superconducting quantum processors, microwave signals are delivered to each qubit from room-temperature electronics to the cryogenic environment through coaxial cables. Limited by the heat load of cabling and the massive cost of electronics, such an architecture is not viable for millions of qubits required for fault-tolerant quantum computing. Monolithic integration of the control electronics and the qubits provides a promising solution, which, however, requires a coherent cryogenic microwave pulse generator that is compatible with superconducting quantum circuits. Here, we report such a signal source driven by digital-like signals, generating pulsed microwave emission with well-controlled phase, intensity, and frequency directly at millikelvin temperatures. We showcase high-fidelity readout of superconducting qubits with the microwave pulse generator. The device demonstrated here has a small footprint, negligible heat load, great flexibility to operate, and is fully compatible with today's superconducting quantum circuits, thus providing an enabling technology for large-scale superconducting quantum computers.

Quantum computing holds the fascinating idea of harvesting computing power from the basic laws of quantum mechanics. The scale of a quantum computer is crucial to empowering noisy intermediate-scale quantum (NISQ) applications and running fully error-corrected quantum algorithms. For superconducting quantum computing, it is believed that millions of qubits are required to achieve fault tolerance[1–4]. How to build such a machine is still an open question. In today's superconducting quantum computer, as illustrated in Fig. 1a, manipulation and readout of qubits require huge costs of generating and routing microwave signals from room temperature to the quantum chips residing in a millikelvin cryogenic dilution fridge. This approach may be extended up to 1000 qubits but is hard to scale further[5]. Monolithic integration of microwave electronics with the qubits, as illustrated in Fig. 1b, provides a promising approach towards the scaling era[5,6]. By replacing the macroscale wiring harnesses and connectors with tightly integrated circuit blocks and chip stackings,

the system's footprint and passive heat load, as the main bottlenecks for scaling, can be dramatically reduced. It would also provide systematic advantages including lower communication latency, improved reliability of IOs, and upgraded signal fan-in/fan-out[6–8].

The major challenge for such integration comes from the requirement of a coherent microwave pulse generator with ultra-small heat load, stringently constrained by the cooling power of the dilution fridges that hold the quantum chip. Emerging technologies utilize cryogenic complementary metal oxide semiconductor (CMOS) circuits[9,10] or photonic links[11–14] to improve the scaling of the quantum computer. However, monolithic integration of those devices with superconducting qubits is challenging due to their strong active heat load[6,9,14]. Superconducting electronics, such as single-flux quantum (SFQ) devices[15–18], can operate at the 10 mK environment but would introduce quasi-particle poisoning to the superconducting qubits that are integrated with SFQ electronics[19]. Josephson junction laser[20] can

[1]Center for Quantum Information, Institute for Interdisciplinary Information Sciences, Tsinghua University, Beijing, PR China. [2]Hefei National Laboratory, Hefei, PR China. [3]These authors contributed equally: Zenghui Bao, Yan Li. ✉e-mail: hyzhang2016@tsinghua.edu.cn; lmduan@tsinghua.edu.cn

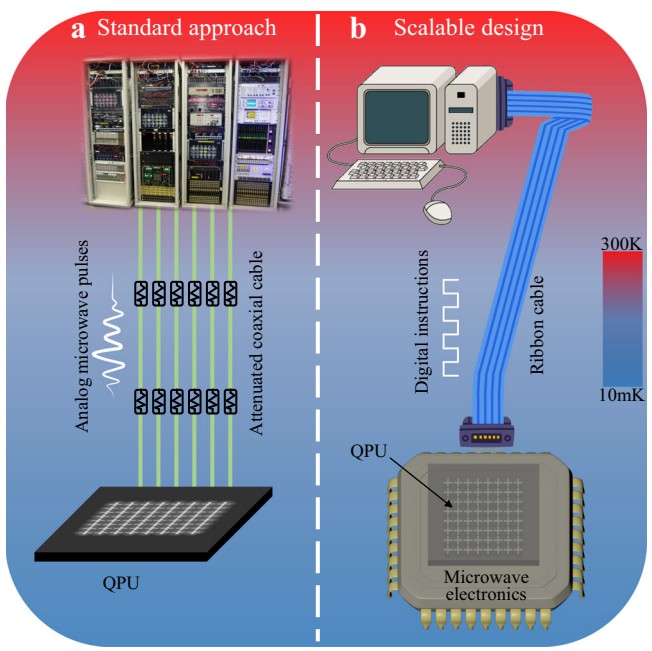

**Fig. 1 | Schematics of superconducting quantum computers. a** The conventional approach to manipulating and readout of a superconducting quantum processor. Room temperature electronics are used as control units to generate analog microwave pulses with a well-defined frequency, amplitude, and phase, which are sent to the cryogenic quantum processing unit (QPU) through coaxial cables with careful attenuation and filtering. The significant hardware overhead limits the scaling of the quantum computer. **b** A conceptual superconducting quantum computer that integrates the QPU with its control units in the cryogenic temperature. The control units may compose cryogenic microwave pulse generators and their control electronics. Such a monolithic integrated architecture enables large-scale superconducting quantum computers[5].

generate high-quality continuous-wave microwave signal but is not sufficient for qubit control due to the lack of full waveform generation[21]. A coherent microwave signal source that is compatible with monolithic integration and cryogenic control of superconducting quantum circuits remains an elusive goal.

In this work, we report such a microwave signal source with excellent coherence. We propose and realize a coherent microwave photon generation from vacuum process with superconducting circuits, where microwave photon pulses with well-controlled phase, intensity, and frequency can be conveniently generated through digital-like control of the magnetic flux across a superconducting quantum interference device (SQUID) embedded in a superconducting resonator. Benefiting from their excellent coherence, the microwave pulses can be conveniently superimposed to create more diverse microwave signals, showing clear advantages compared with previously demonstrated microwave photon sources operating in the cryogenic enviroment[20,21]. Prominently, the digital-like signal used to drive the microwave source can be delivered through channels with low bandwidth, such as twisted pairs of wires, which are of a much lower passive heat load than that of the coaxial cables used in today's technology[22]. We further showcase using the microwave pulse generator for high-fidelity qubit state readout. The device demonstrated here is simple, highly scalable, and compatible with superconducting quantum circuits, which would enable monolithic integration of the control electronics and the superconducting qubit at milli-kelvin temperature, thus providing a prime building block for a large-scale superconducting quantum computer.

## Results
### Device design
Our approach is based on engineering photon generation from a vacuum with a SQUID-embedded coplanar waveguide (CPW)

resonator. The magnetic flux across the SQUID periodically changes the effective inductance of the SQUID, and thus the resonator frequency[23,24]. It has been demonstrated that correlated photon pairs with thermal-like spectrum can be generated from the initial vacuum through a fast modulation of the resonator frequency[25], which is attributed to the dynamical Casimir effect, or more generally, a quantum vacuum amplification[26]. To have a coherent state output, we consider a toy model (Supplementary Note 1) that describes the photon generation from an abrupt change in the magnetic flux, resulting in an effectively modified Hilbert space. In this case, the resonator vacuum in the original Hilbert space remains unchanged, while it is a displaced one in the modified space, thus a coherent state. This is analogous to abruptly shifting the suspension point of a pendulum as illustrated in Fig. 2a. This modification can be realized by adding a tunable $a + a^\dagger$ term in the harmonic oscillator Hamiltonian, where $a$ ($a^\dagger$) is the creation (annihilation) operator of the resonator mode. We note that using an asymmetric dc-SQUID design in the resonator can introduce such a term[27], whose coefficient can be rapidly tuned by the magnetic flux, resulting in the change of the system's equilibrium. Importantly, owing to the strong nonlinearity of the SQUID, a significant displacement, thus a large photon number, is expected when the flux goes across the boundary of adjacent periods $(n + 1/2)\phi_0$, where $n$ is an integer and $\phi_0$ is the flux quantum.

In the experiment, the resonator is made from an aluminum thin film on a sapphire substrate. As illustrated in Fig. 2b, it is composed of a $\lambda/2$ CPW resonator with a SQUID embedded at the electric field node of the fundamental mode. The SQUID consists of two Josephson junctions of different areas in parallel. We note that the unbalance of the SQUID is critically influenced by the fabrication uncertainty, especially for a small junction area. The electrical current on the nearby flux line generates magnetic flux across the SQUID. The device is cooled down to about 10 mK in a dilution fridge (Supplementary Fig. 6). The fundamental mode frequency of the resonator shows periodical dependence on the applied flux $\phi$, as shown by the three branches in Fig. 2c. At $\phi = n\phi_0$, the resonator frequency is maximized as sweet spots, whereas when $\phi = (n + 1/2)\phi_0$, the resonator frequency approaches a minimum.

### Microwave emission
We first apply a step magnetic field to the SQUID, for which the end magnetic flux is fixed at the left frequency branch in Fig. 2c while the initial flux is scanned within a broad range. As predicted by the toy model, if the step magnetic flux $\phi$ changes across the odd multiples of the half flux quantum $(n + 1/2)\phi_0$, numerous microwave photons can be generated in the CPW resonator and subsequently leak out to the transmission line. Such a phenomenon is explicitly displayed in Fig. 2e, where the output photon number is recorded as a function of the initial flux. The microwave photon number is calibrated based on the measurement-induced dephasing of a superconducting qubit (Supplementary Note 3). Figure 2d shows a typical time series of the quadratures after down-converting the emitted microwave signal to tens of megahertz, which contains pulsed coherent state signals with a well-defined phase and frequency and an exponentially decreased envelope. The decay time constant of the pulse signal is mainly determined by the linewidth of the resonator (see Supplementary Fig. 7). We note that tuning the envelope of the microwave pulse to realize the desired waveform is possible by introducing a tunable out-coupling of the signal source to the external circuits[28,29]. Importantly, the phase of the coherent output can be continuously tuned from 0 to $2\pi$ by controlling the initial value of the magnetic flux step, as shown in Fig. 2f. This is of great importance for quantum device manipulation where the phase of the microwave signal matters. Such an intrinsic coherence of the microwave emission is distinctly different from the previously reported microwave signal generation effect in cryogenic temperature[20,21].

Crucially, the photon number and frequency of the microwave emission can be conveniently tuned with the applied flux step. Figure 2g

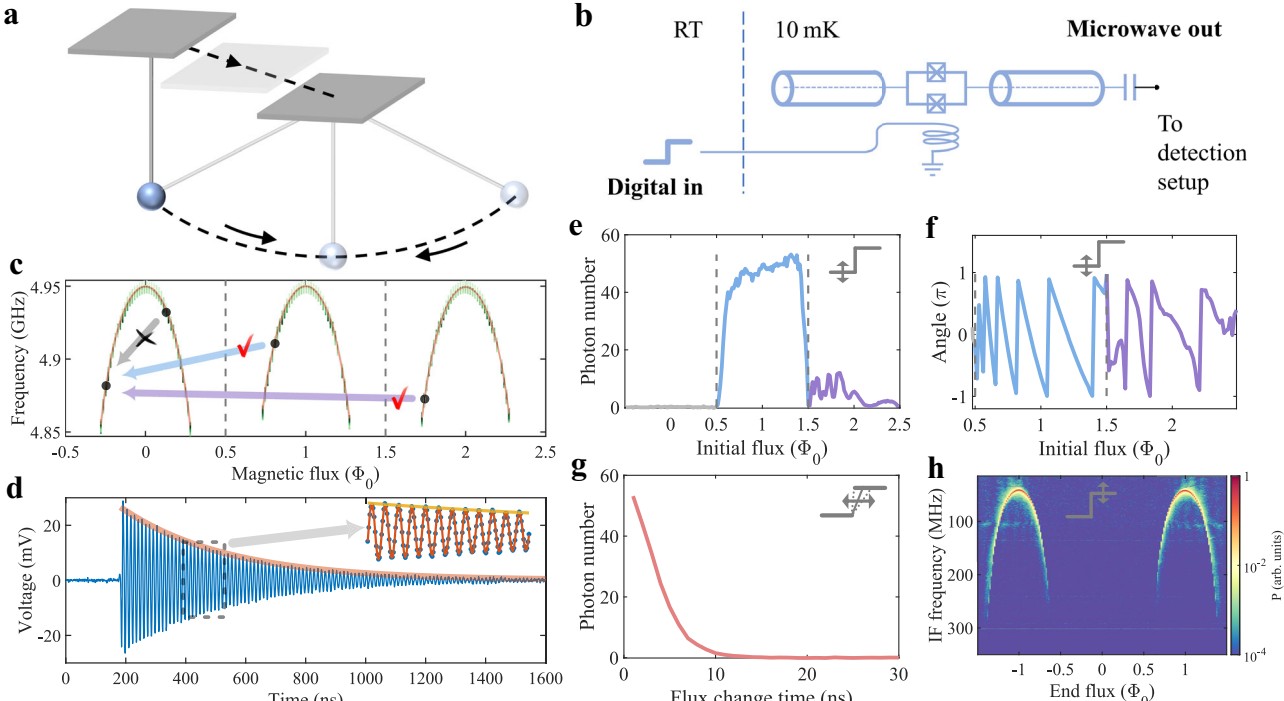

**Fig. 2 | A coherent microwave pulse generator. a** Pendulum oscillation induced by abruptly shifting the suspension point. During the sufficiently fast move, the pendulum globe almost keeps still. However, it is no longer at the equilibrium position of the modified pendulum, thus oscillation occurs thereafter. **b** Circuit diagram of the cryogenic microwave pulse generator, composing a $\lambda/2$ coplanar waveguide (CPW) resonator with a superconducting quantum interference device (SQUID) embedded in the middle of the center conductor. A flux line is used to control magnetic flux across the SQUID, and thus operate the pulse generator. **c** The measured resonance frequency of the resonator shows a periodic dependence on the magnetic flux across the SQUID, which can be well-fitted with a theoretical curve (orange lines)[23,24]. As illustrated by the arrowed lines, by applying a fast change of magnetic flux across the $\phi = (n+1/2)\phi_0$ point (gray dashed lines),

where $n$ is an integer and $\phi_0$ is the flux quantum, microwave pulses can be generated in the resonator and released to the measurement setup. **d** The emitted microwave pulse has an exponential decay envelope (orange line) with a time constant determined by the linewidth of the resonator. The inset of **d** is a zoom-in of the time series, which can be well-fitted with an exponential-enveloped sinusoidal function (red line), exhibiting well-defined coherence. **e** Output photon number from the pulse generator driven with a varied initial flux, showing that the flux change across at least one $(n+1/2)\phi_0$ point is necessary to generate microwave pulse. **f**–**h** the phase, photon number, and frequency of the microwave pulse can be well controlled by tuning the starting flux, slope, and ending flux of the flux steps used to operate the pulse generator, correspondingly. The insets of **e**–**h** illustrate the applied magnetic flux steps.

illustrates that the photon number of the generated microwave pulse can be well-controlled by fixing the initial and end flux while using different flux change rates. In the experiment, the maximum photon number generated in a single microwave pulse is about 1000 when the flux step is generated with a 1 GHz sampling rate, which can be further improved by optimizing the device parameters and increasing the change rate of the flux step (Supplementary Note 2). The frequency of the emission is determined by the frequency of the resonator and thus can be controlled by the end flux of the signal step. In Fig. 2h, we fix the initial flux to the central frequency branch and scan the end flux, showing that the frequency of the microwave pulse can be continuously tuned by more than 200 MHz. Note that the output photon number shows a clear decrease when the frequency is detuned from the sweet spot, which is mainly due to the increased internal loss rate, and can be mitigated by using Josephson junctions with smaller transparency (Supplementary Fig. 8)[23]. Meanwhile, by increasing the out-coupling rate of the signal source to the external circuits, the generated microwave photons can leak out faster, thus achieving stronger emission power. It is worth noting that the phase and power tunability can be well-reproduced through the numerical simulation of the toy model based on the circuit model (Supplementary Note 2).

### Versatile control

From the toy model, photon generation requires only a fast passing through the boundary of adjacent frequency periods. Therefore, we anticipate that the microwave source can be driven by various forms of

magnetic flux other than a step function. As an example, we use a $\delta$ function-like magnetic flux pulse with an overshoot across a threshold to trigger the microwave emission. In this case, the frequency of the emission is determined by the base flux. The phase and output photon number can be tuned by the overshoot and the width of the flux pulse, respectively. The results are shown in Supplementary Fig. 10.

Remarkably, benefiting from the relatively small bandwidth of the digital drive signal, twisted pairs of wires can be used to deliver the step drive signal from room temperature to cryogenic temperature to actuate the microwave emission. Even though the flux drive signal is distorted by the limited bandwidth of the twisted pairs, well-controlled microwave pulses can still be effectively generated, but with a lower output power, which is typically about 0.07 times that of the coaxial cables and can be improved by optimizing the design of the twisted pairs[30]. Detailed results can be found in Supplementary Note 4.

### Superimposed emission

Taking advantage of the well-controlled phase coherence of the microwave generation process, one could conveniently superimpose outputs from different microwave sources, or multiple microwave pulses from the same source as desired. In Fig. 3a, we connect two cryogenic sources and prepare microwave emissions simultaneously by driving them with flux steps. By tuning the respective end fluxes, the generated microwave pulses can be aligned to the same frequency and superimposed at their common output. Figure 3b shows that by varying the initial flux, and thus the phase of the microwave pulse for

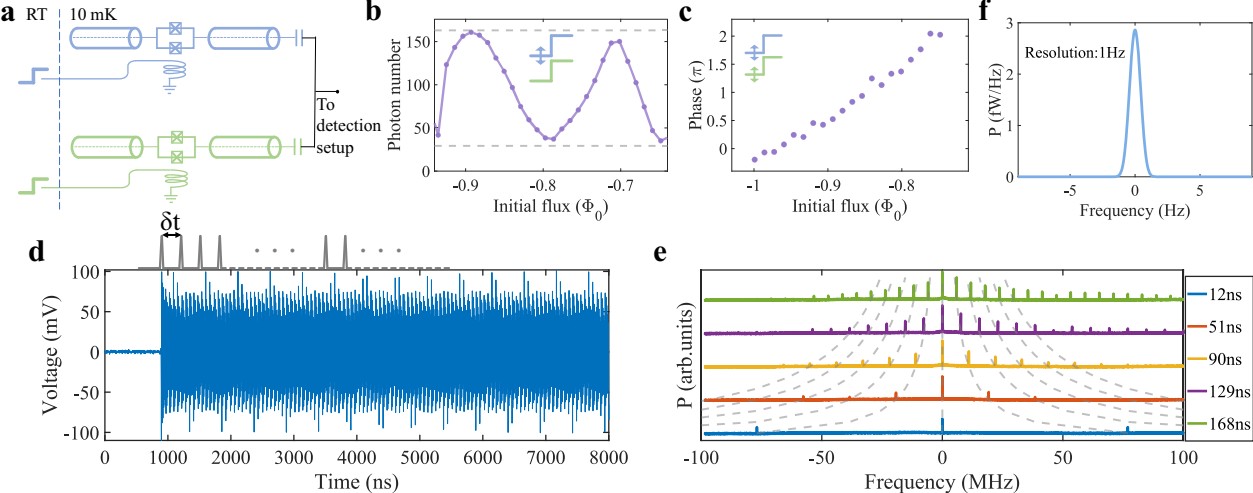

**Fig. 3 | Superimposed microwave emission. a** Circuit diagram used for superimposing two microwave pulse generators. **b** By altering the initial value of the flux step, thus the phase of the microwave pulse for one of the signal sources, the photon number of the superimposed output can be periodically tuned between a constructive interference and a destructive interference, as indicated by the dashed gray lines. Those upper and lower limits are inferred based on the respective output photon number from the two devices and the temporal mode mismatch between the pulses. **c** The phase of the superimposed signal can also be tuned while keeping the output energy at the maximum. This is realized by using suitable combinations of the initial fluxes for the two signal sources to constructively superimpose the two pulses with the given phase. **d** Time series of a continuous-wave (CW) microwave signal, which is prepared by using a train of $\delta$ function-like magnetic flux pulses to operate the microwave source. **e** The frequency spectrum of a typical CW emission is a frequency comb. The frequency spacing of the comb can be tuned by the interval of the applied flux pulses $\delta t$. The dashed line is given by $n/\delta t$, where $n$ is an integer. The power spectral densities have been normalized and offset for comparison purposes. **f** A typical linewidth of the CW emission measured with a spectrum analyzer, showing a resolution-limited linewidth below 1 Hz. Details about the linewidth estimation can be found in "Methods" section and Supplementary Fig. 9.

one of the sources, the photon number of the superimposed output shows oscillation between that for constructive interference and destructive interference. Note that for a given emission energy, one could control the phase of the superimposed output by first choosing a suitable initial flux for one device that corresponds to the desired phase, and then scanning the initial flux for the other device to reach the aimed emission energy. As an illustration, in Fig. 3c the phase of the superimposed signal is continuously tuned while the output energy is maintained at the maximum. Those results demonstrate the remarkable scalability of such microwave photon sources to generate microwave signals with tunable properties.

It is also possible to superimpose microwave pulses from the same signal source. As illustrated in the top panel of Fig. 3d, we use a train of flux overshoots with a time interval of $\delta_t$ as a drive to generate multiple microwave pulses with the same initial phase at varied delays. The interference of those pulses results in a continuous-wave (CW) microwave output in the form of a frequency comb, as shown in Fig. 3d and e. The comb spacing can be well-controlled by the time interval $\delta_t$. Especially, when $\delta_t \times f$ is an integer, where $f$ is the frequency of the microwave emission, continuous microwave output with maximized sideband suppression can be prepared. Such a flexible frequency comb could be used in many superconducting-circuit-based quantum information processing tasks, including multiplexed quantum measurement[31] and optimized quantum control[32]. Prominently, the linewidth of those frequency components can be narrower than 1 Hz without any injection locking, as demonstrated in Fig. 3f. A detailed lineshape analysis assigns a linewidth on the order of 1 mHz, as discussed in the Methods and Supplementary Fig. 9. Those results imply a high-frequency resolution and stability of the microwave source, which is important for a variety of technologies including sensing and spectroscopy[33–35].

## Qubit readout

The cryogenic microwave pulse generator can be used for the readout of superconducting qubits. In Fig. 4a, we prepare microwave pulses with flux steps and send them to a superconducting resonator that is dispersively coupled to a transmon qubit[36]. The reflected signal is collected and analyzed with a homodyne setup. By varying the end magnetic flux used to drive the pulse generator, and thus the frequency of the output signal, the qubit-state-dependent resonator responses can be measured with the cryogenic source, as shown in Fig. 4b, which is similar to that recorded with the conventional method[37]. To demonstrate qubit state readout with the microwave source[38], we place the microwave pulse frequency to that with the maximum contrast for the qubit-state-dependent resonator responses and continuously rotate the qubit from the ground state $|g\rangle$ to the excited state $|e\rangle$. The phase of the reflected signal is presented in Fig. 4c, showing that the qubit state can be effectively read out with the cryogenic signal source.

Single-shot readout of the qubit state is necessary for implementing many quantum algorithms and realizing quantum error correction[39,40]. From the microwave source, we generate a single microwave pulse or two consecutive ones (Fig. 4d) with approximately 24.0 photons or 41.8 photons, correspondingly, and send them to the readout resonator. With the qubit initialized in $|g\rangle$ or $|e\rangle$, we record the histograms of $5 \times 10^4$ measurements of the homodyne signals that take the temporal mode of the microwave pulses into account. As shown in the inset of Fig. 4d, well-separated Gaussian distributions corresponding to $|g\rangle$ or $|e\rangle$ state of the qubit can be resolved. We extract single-shot readout fidelities of $97.29 \pm 0.13\%$ for a single microwave pulse, and $97.85 \pm 0.14\%$ for two pulses, identical to the fidelity of $97.78 \pm 0.33\%$ obtained with the conventional method (2 $\mu$s square pulse, approximately 25.5 photons).

## Qubit drive

We consider two approaches to perform the single qubit operation with the cryogenic microwave source and theoretically estimate the possibly achieved single qubit gate speed, respectively, as shown in Supplementary Fig. 11. Similar to the conventional method, we can connect the output of the signal source to the qubit through an open

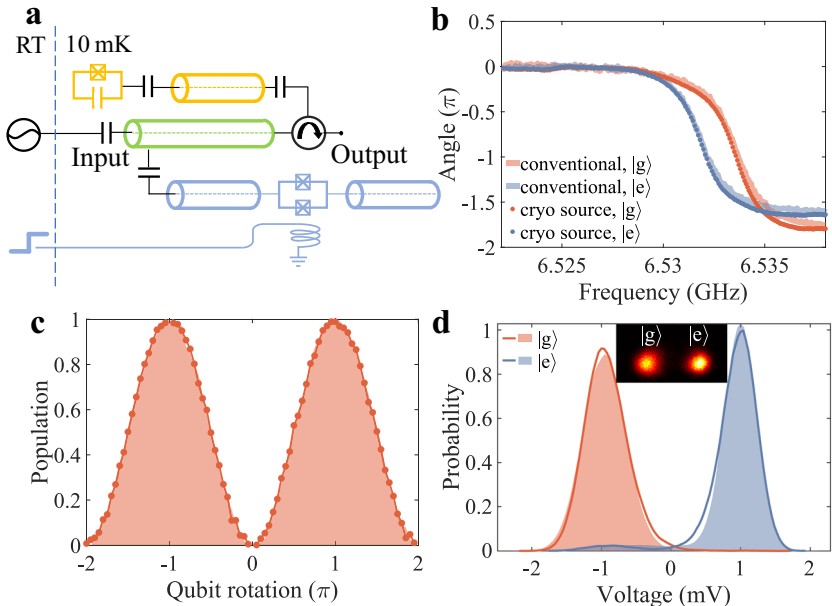

**Fig. 4 | Qubit readout with the cryogenic pulse generator. a** Circuit diagram for the readout of qubit state with the cryogenic pulse generator (blue). The qubit sample (orange) is implemented with a 3D circuit quantum electrodynamics architecture containing a transmon qubit that is dispersively coupled to a microwave resonator for state readout. The conventional readout signal is synthesized with room-temperature electronics as introduced in "Methods" section and Supplementary Fig. 6. A coplanar waveguide (green) is used to connect both the cryogenic pulse generator and the conventional readout signal to the qubit sample.

**b** Qubit-state-dependent reflection spectra of the readout resonator measured with both the conventional method and the pre-calibrated microwave pulses from the cryogenic signal source. **c** Rabi oscillation of the transmon qubit measured with the conventional method (shadow) and the cryogenic pulse generator (scattered plots). **d** Histograms of the resonator response measured the conventional method (shadow) and the cryogenic pulse generator (scattered plots), with the qubit prepared in $|g\rangle$ and $|e\rangle$. The inset shows a typical distribution in the I-Q quadrature plane measured with the cryogenic pulse generator.

waveguide. In this case, the Rabi rate that can be achieved is on the order of 0.26 MHz, with the maximum output power obtained so far and reasonable coupling strength between the waveguide and the qubit, which is still too slow for practical superconducting quantum computing. Alternatively, we can directly connect the resonator of the signal source to the qubit and tune them on resonance when performing the qubit gate operation. With a similar coupling strength and photon numbers, the estimated Rabi rate is about 30 MHz, which is much faster than that of the conventional method and agrees well with the previous reports[41,42]. We note that such an 'energy-saving gate' is preferred for large-scale quantum computing[42]. In addition, the same resonator can be used for qubit state readout and reset[43], which leads to a more compact design.

A stronger output power is preferred to realize fast and high-fidelity qubit state manipulation. A direct approach is to superimpose multiple microwave pulses to harvest more microwave photons with either multiple signal sources or using a series of flux steps/overshoots as drive, as demonstrated in Fig. 3b–d. Additionally, from our simulation results, stronger emission of a single microwave pulse can be obtained by increasing the flux change rate and fine-tuning the symmetry of the resistances of the Josephson junctions in the SQUID. A detailed discussion can be found in Supplementary Note 2.

## Discussion

In this work, we have realized a coherent on-chip cryogenic microwave pulse generator with negligible heat load and simple control, which provides distinct benefits for large-scale superconducting quantum computing. Benefiting from the small footprint and the compatibility with superconducting qubits, the pulse generator can be monolithically integrated with superconducting quantum circuits without using macroscale wiring harnesses and connectors, which will greatly improve the density and reliability of the input-output packaging[6–8].

Replacing the expensive microwave electronics and devices with chip-scale integrated circuits will also provide great economic feasibility to large-scale superconducting quantum computers. Moreover, compared with the conventional method, microwave pulses directly generated at the millikelvin temperatures can be sent to the qubits without the need for attenuation to suppress blackbody radiation from higher temperatures, which lessens a considerable amount of heat load at the millikelvin region, especially when operating a large number of qubits[14,22,42].

In the current prototypical demonstration, coaxial cables are used to deliver the driver signal of the pulse generator. We note that given the small bandwidth requirement, the pulse generator can be actuated even through superconducting twisted pairs of wires, for which the passive heat load is several orders of magnitude smaller than that of the coaxial cables[22]. Moreover, deep cryogenic devices that are capable of generating the signal edges or voltage pulses required for the operation of the microwave source have been demonstrated before[9,17]. Therefore, we anticipate that it may be possible to compactly integrate the microwave pulse generators demonstrated here and their controlling circuitries directly in the cryogenic environment, which functions as the control units for the quantum processor and can be controlled with digital instructions provided by a small number of wires connected to the room temperature electronics, as illustrated in Fig. 1b. Such a tight integration would lift a major roadblock toward large-scale superconducting quantum computation[6].

## Methods
### Sample information

The cryogenic microwave pulse generator is constructed with a $\lambda/2$ coplanar waveguide (CPW) resonator and a superconducting quantum interference device (SQUID) embedded in the center conductor and placed at the electric field node of the fundamental mode. The room temperature junction resistances used in the experiment range from

50Ω to 270Ω, effectively corresponding to about 58 pH to 310 pH at the zero flux point, which occupies about 3.1–11.6% of the total inductance of the SQUID-embedded resonators. The Inset of Supplementary Fig. 6 shows a photomicrograph of a typical cryogenic coherent microwave photon source used in the experiment.

The SQUID consists of two parallel Josephson junctions and functions as a tunable inductor $L_j(\Phi_{ext}) = \Phi_0/[2\pi I_c \cos(\pi\Phi_{ext}/\Phi_0)]$ for the symmetric junctions case, where $I_c$ is the critical current of the junction and $\Phi_0 = h/2e$ is the flux quantum. The inductance $L_j(\Phi_{ext})$ can be controlled by the magnetic flux $\Phi_{ext}$ through the SQUID loop generated by the electrical current on the nearby flux line. The total inductance of the SQUID-embedded resonator comprises both the inductance of the CPW resonator $L_r$ and the flux-dependent SQUID inductance $L_j(\Phi_{ext})$. Consequently, the relation between the resonance frequency of the fundamental mode $\omega$ and the loop flux $\Phi_{ext}$ can be determined as

$$\omega = \sqrt{\frac{1}{(L_r + L_j(\Phi_{ext}))C_r}} \tag{1}$$

where $C_r$ is the capacitance of the CPW resonator and the capacitance of SQUID is small enough to be neglected. Eq. (1) is used to fit the experimental result shown in Fig. 2b.

The qubit sample used for the readout experiment is implemented with a 3D circuit quantum electrodynamics architecture. A superconducting transmon qubit is patterned on a 7 mm × 7 mm sapphire substrate with standard micro-fabrication techniques. The Josephson junction of the qubit is fabricated with electron-beam lithography and double-angle evaporation of aluminum. The qubit chip is placed in and dispersively coupled to a 3D aluminum resonator. Detailed parameters of the qubit device can be found in Supplementary Table 1.

## Experimental setup

The measurement setup is schematically shown in Supplementary Fig. 6.

The flux step or overshoot used to drive the pulse generator is generated with an arbitrary waveform generator (AWG) with a 1 GHz sampling rate, and delivered through either coaxial cables or superconducting twisted-pair wires to the signal source located at the 10 mK region. The output of the cryogenic microwave source is successively amplified by a high-electron-mobility transistor (HEMT) amplifier at 4-K plate and two microwave amplifiers at room temperature. A spectrum analyzer is used to characterize the spectra and linewidth of the CW output from the cryogenic signal source. A homemade homodyne setup is used to analyze the in-phase and quadrature components of the output signal, thus extracting the amplitude and phase of the corresponding output. In the homodyne setup, a commercial microwave signal source is used as a local oscillator (LO) to down-convert the amplified gigahertz signals to tens of megahertz signals, which are later amplified by a voltage amplifier and acquired by an analog-to-digital converter (ADC) with a 1GHz sampling rate. It is worth noting that the repetition rate for the data acquisition is carefully chosen to make sure the phase of the LO signal is the same in every single trial of the experiments.

In the qubit readout experiment, the microwave pulses used for qubit control and conventional readout are prepared and delivered to the qubit sample with the conventional method. To be specified, the microwave pulses are synthesized by modulating the continuous-wave (CW) gigahertz carrier signal from a commercial signal generator with a megahertz signal from an AWG, which is used to control the amplitude and phase of the microwave pulses. The synthesized microwave pulses are sent to the qubit sample through coaxial cables with proper attenuation and filtering to suppress possible noises. When measuring with the cryogenic microwave source, the output of the signal source is

directed to the transmon qubit sample. To determine the qubit state, the reflected signal from the qubit readout resonator is sent to the amplifier chain started with a Josephson parametric amplifier. The amplified signal is analyzed with the homodyne setup.

## Generation of microwave frequency comb

As mentioned in the main text, with a train of flux overshoots as a drive, microwave pulses with the same initial phase can be superimposed to generate CW microwave signals. To understand the spectral information of the CW signal, we take a simplified model by directly adding the truncated ideal time series of the microwave pulses as

$$f(t) = Ae^{-\frac{\gamma}{2}(t-n\delta_t)}\sin(\omega_e(t-n\delta_t)), n\delta_t < t < (n+1)\delta_t \tag{2}$$

where $n$ takes all integers. $\delta_t$ is the time interval between the repetitive flux overshoots. $\omega_e$, $A$ and $\gamma$ are the frequency, amplitude, and decay time constant of the microwave pulses, respectively. The ideal periodic function can be expanded into Fourier series $f(t) = \sum_{n=-\infty}^{n=+\infty} c_n e^{in\delta\omega t}$, which is a frequency comb centered at $\omega_e$. The expansion coefficient can be derived as

$$c_n = \frac{A}{\delta_t} \frac{\omega_e + e^{-\Gamma\delta_t}(i\omega_e \cos(\omega_e\delta_t) - \Gamma\sin(\omega_e\delta_t))}{\Gamma^2 + \omega_e^2} \tag{3}$$

where $\delta\omega = 2\pi/\delta_t$ is the spacing of the frequency comb, $\Gamma = (\frac{\gamma}{2} + in\delta\omega)$. It explains the results shown in Fig. 3d that the comb spacing is inversely proportional to the time interval $\delta_t$.

To have a single-color microwave emission at $\omega_e$, or to maximize the suppression ratio between the principal maximum and the sidebands, the time interval $\delta_t$ is set to an integer multiple of the microwave signal period to meet $\omega_e \times \delta_t = 2n'\pi$, where $n'$ is an integer. In this situation the the endpoint of the last pulse can in principle match the starting point of the following pulse, and the expansion coefficient can be simplified as

$$c_n = \frac{A}{\delta_t} \frac{\omega_e + i\omega_e e^{-\Gamma\delta_t}}{\Gamma^2 + \omega_e^2} \tag{4}$$

One can find that the strengths of the sideband components are determined by both the time interval $\delta_t$ and the time constant $\gamma$ of the exponential envelope. In the experiment, we achieve a sideband suppression ratio of about 24.8 dB for $\omega_e/2\pi = 6.5401$ GHz with 1 ns time resolution of the AWG. This can be further improved by applying flux overshoots with better time resolution and using signal sources with smaller $\gamma$.

## Linewidth estimation for the CW signal

The CW microwave signal generated by the cryogenic source is analyzed with a spectrum analyzer (R&S FSV3030) to learn the spectrum information. As shown in Supplementary Fig. 6, the output signal is amplified by the amplifier chain and sent to the spectrum analyzer, which is set to the minimum resolution bandwidth (RBW) of 1 Hz. The power spectrum density of the CW signal centered at its principal maximum is shown in Supplementary Fig. 9. The measured data manifests a Voigt line shape, which is dominated by a Gaussian function with a full width at half maximum of ~0.93 Hz. This Gaussian function of about 1 Hz bandwidth is introduced by the limited resolution of the spectrum analyzer. Since a Gaussian-like filter is used to analyze the input signal, the measured power spectrum density is a convolution of the intrinsic spectrum of the input signal and the Gaussian filter. It means that the linewidth of the input signal can not be clearly distinguished below the minimum RBW of the given setup. The Voigt line shape of the measured data indicates the intrinsic linewidth of the CW signal is much narrower than the minimum RBW of

1 Hz. We can estimate a bound on the linewidth of the CW signal from the measured data.

Assuming the CW signal has a Lorentzian line shape with a certain full width at half maximum (FWHM), the convolution of a Gaussian function and Lorentzian function gives a standard Voigt line shape, which features wing-like upward tilts on both sides of the center frequency. These tilts are determined by the FWHM of the Lorentzian. The narrower (broader) Lorentzian results in a farther (closer) inflection point from the center frequency accompanied by a lower (higher) power. According to the position of the inflection point, we can extrapolate the bound of the signal linewidth below the minimum RBW 1 Hz of the spectrum analyzer by varying the FWHM of the Lorentzian component of the convolution. The tilts of the experimental data are located in between the 2 mHz case and the 0.5 mHz case, and the inflection point is slightly below the 0.5 mHz case. The result indicates the linewidth of the CW signal is on the order of 1mHz. Notably, The wing regions of the experimental data are different from an ideal Voigt lineshape, which is attributed to the non-ideal Gaussian filter in the spectrum analyzer.

## Qubit state readout with the cryogenic microwave source

The experimental setup to measure the qubit state with the cryogenic microwave source is illustrated in Supplementary Fig. 6. The transmon qubit consists of a single Josephson junction shunted by a capacitor, which is dispersively coupled to a 3D microwave resonator used for qubit state readout. Detailed parameters about the qubit sample can be found in Supplementary Table 1. The microwave signal generated by the cryogenic source is sent to the readout resonator through a circulator, with which the reflected signal by the resonator can be measured. The reflection signal is amplified and analyzed with a homodyne setup.

To measure the reflection spectra of the readout resonator with the cryogenic signal source (Fig. 4a), we have to first calibrate the output frequency of the signal source for different end fluxes, which is similar to the measurement shown in Fig. 2g. Besides the frequency dependence on the end flux, the emission intensity and phase with varied end fluxes are also recorded as background signals. In the experiment, we use different end fluxes to drive the signal source and send the output to the readout resonator. The reflected signal is measured as raw data, from which the background is substrated to extract the real resonator response in Fig. 4a.

Notably, to achieve maximized detection efficiency, the temporal mode of the microwave pulse generated by the signal source has to be considered, especially for the single-shot readout of the qubit state. The output microwave pulse can be depicted by the time-independent mode $a$ expressed as a function of the time-dependent field mode $a_{out}$ as

$$a = \int dt\, f(t) a_{out}(t) \qquad (5)$$

where $f(t)$ represents the temporal mode function which satisfies the normalization condition $\int dt\, |f(t)|^2 = 1$ and guarantees the commutation relation $[a, a^\dagger] = 1$. The time-dependent operator $a_{out}$ corresponds to the time trace of the voltage recorded by the ADC, and the time-independent operator $a$ corresponds to the in-phase (I) and quadrature (Q) moments extracted from the time trace with a digital homodyne method[44]. By setting a proper envelope for the digital homodyne function, which is the same as the envelope of the recorded time trace, the digital homodyne process can reach unity efficiency, and the moment information can be fully extracted from the time trace. In the case of conventional readout, considering that the readout pulses are usually prepared with a square-shaped envelope and only slightly distorted by the resonator, a square shape is used as the envelope for the digital homodyne.

Since the microwave pulse generated by the cryogenic source has an exponential decay envelope, using a square wave leads to

reduced detection efficiency. Additionally, when the microwave emission pulse is used to read the qubit state, the envelopes of the reflected signals when the qubit is in $|g\rangle$ and $|e\rangle$ can be very different due to the dispersive-shifted resonance of the readout resonator. Therefore, in the single-shot experiments, we employ the averaged readout signals when the qubit is in $|g\rangle$ and $|e\rangle$ as the digital homodyne functions, respectively, instead of using one fixed envelope. Accordingly, the effective quadratures I and Q can be extracted as

$$I = \int dt\, f_g(t) V(t) \qquad (6)$$

$$Q = \frac{\int dt\, f_e(t) V(t) - I \cos\theta}{\sin\theta} \qquad (7)$$

where $f_g(t)$ ($f_e(t)$) is the normalized average output signal when the qubit is in $|g\rangle$ ($|e\rangle$). $V(t)$ represents a single trail output signal in the single-shot readout experiment; $\theta = \arccos(\int dt\, f_g(t) f_e(t))$ is the overlap angle between the two average signals corresponding to the two qubit states. It is worth noting that the average output signals $f_g$ and $f_e$ driven by the exponentially-enveloped input are non-orthogonal. When using two non-orthogonal signals as digital homodyne functions, one of the two calculated quadrature moments needs to be compensated for the overlap angle. Here, $f_g(t)$ is taken as an eigenvector of the spanned I-Q space to give the in-phase moment, and the quadrature moment is compensated as shown in Eq. (7). By using this temporal mode fitting method, the moment information carried by the resonator reflection can be efficiently extracted to distinguish the qubit states in single-shot experiments.

## Qubit drive estimation

We consider two approaches to perform a single-qubit gate with the microwave source, and estimate the possibly achieved single-qubit Rabi rate.

The first approach is similar to the conventional qubit drive scheme, for which the output of the signal source is coupled to the qubit via an open waveguide. Taking the coupling strength between the waveguide and the qubit as $\Gamma_{ext}$, the Rabi rate $\Omega$ can be estimated as

$$\Omega = 2\sqrt{\dot{n}_d \Gamma_{ext}}, \qquad (8)$$

where $\dot{n}_d$ is the photon flux in the waveguide. Considering the energy consumption during the single qubit $\pi$ rotation, the Rabi rate can be expressed as the function of total input photon number $n$ in the waveguide

$$\Omega = \frac{4}{\pi} n \Gamma_{ext}. \qquad (9)$$

Here, we take a reasonable coupling strength $\Gamma_{ext} = 1000$ Hz for calculation, corresponding to a qubit lifetime limitation 1ms, and the result is shown in Supplementary Fig. 11 with the purple line. According to the maximum output power obtained with the cryogenic microwave source, the achievable Rabi rate is about 0.26MHz, which is too slow compared with the practical qubit gate speed implying an insufficient output power.

In the second approach, the resonator of the signal source is directly connected to the qubit through a capacitive coupling. The coupling capacitance is chosen to be the same as that between the waveguide and the qubit in the first approach. The resonator and the qubit are tuned on resonance when performing the gate operation and tuned far off-resonance when idle. To simulate the single qubit gate,

we consider the Jaynes-Cummings Hamiltonian

$$\hat{H} = \frac{\omega_q}{2}\hat{\sigma}_z + \omega_r\hat{a}^\dagger\hat{a} + g\left(\hat{a}^\dagger\hat{\sigma}_- + \hat{a}\hat{\sigma}_+\right), \qquad (10)$$

where $\hat{a}$ ($\hat{a}^\dagger$) is the annihilation (creation) operator of the resonator mode, and $\hat{\sigma}$ is the Pauli operator of the qubit mode; $\omega_q$ and $\omega_r$ are the frequency of the qubit and resonator of the cryogenic signal source, respectively; $g$ is the coupling strength between the qubit and the resonator. Since the coupling strengths $\Gamma_{ext}$ and $g$ in the two approaches are both determined by the coupling capacitance $C_c$, it is intuitive to compare the Rabi rates of the two approaches under the same coupling capacitance. According to a simple capacitive coupling model, the relations between the two coupling strengths and the coupling capacitance can be expressed as

$$C_c = \sqrt{\frac{C_q\Gamma_{ext}}{\omega_q^2 Z_0}},$$
$$g = \frac{C_c}{2C_qC_r}\sqrt{\frac{1}{Z_rZ_q}}, \qquad (11)$$

where $C_q$ and $C_r$ are the capacitance of the qubit and the resonator; $Z_0$, $Z_q$, and $Z_r$ are the characteristic impedances of the open waveguide, the qubit, and the resonator. With a common qubit parameter setting $\omega_q = 6$ GHz, $E_c = 220$ MHz, the coupling strength $\Gamma_{ext} = 1000$ corresponds to the coupling capacitance $C_c = 35$ aF. Therefore, the coupling strength in the second approach can be determined as $g = 0.49$ MHz. By considering the coherent states of different photon numbers in the resonator, the simulation results of the related Rabi rate are shown in Supplementary Fig. 11 with the pink line. In this approach, the Rabi rate is much faster than the conventional method with the same photon number. The pink triangle and square indicate the Rabi rates 30.8 MHz and 8.23 MHz corresponding to the maximum photon number experimentally generated in the resonator 1017 and 71 when the flux drive signal of the cryogenic microwave source is delivered through the coaxial cables or twisted-pair wires. By optimizing the twist pitch length of the pair of wires and thus their bandwidth[30], the maximum achieved photon number is estimated to be 662, leading to the optimized Rabi rate of 24.9 MHz, as shown with the pink diamond in Supplementary Fig. 11.

## Data availability
The data that support the findings of this study are available from the authors upon request.

## Code availability
The codes of this study are available from the corresponding authors upon request.

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

## Acknowledgements

This work was supported by the Innovation Program for Quantum Science and Technology (2021ZD0301704), the Tsinghua University Initiative Scientific Research Program, the Ministry of Education of China, and the National Natural Science Foundation of China under Grant Nos. 12374472 and 92165209.

## Author contributions

H.Z. and L.D. supervised the project. H.Z. conceived the idea and proposed the experiment. Z.B. and Y.L. prepared the sample and collected the data with the assistance of Z.W., J.W., J.Y., H.X., and Y.S.; H.Z. and Z.B. analyzed the data. Z.B., H.Z., and Y.W. carried out the theoretical model. H.Z. and Z.B. wrote the manuscript. All authors contributed to the discussions and production of the manuscript.

## Competing interests

The authors declare no competing interests.
