## [Peer Review File · Nature Communications]

REVIEWER COMMENTS

Reviewer #1 (Remarks to the Author):

In the manuscript entitled "A cryogenic on-chip microwave pulse generator for large-scale superconducting quantum computing", Zenghui Bao et al., reported the realization of a compact on-chip pulse generator, different from the more well-established SFQ device, and demonstrated that the pulse generator can potentially meet the requirement of driving superconducting qubit.

It is an interesting technical paper. The topic, mitigating the scalability problem in a large-scale quantum computing system, is certainly of broad interest. However, I am not convinced that the work presents the level of novelty that warrants the publication in Nature Communications.

Major issue:

The major issue is, in my view, what is the advantage of this work? After all, the number of required pulse generator scales linearly with the number of qubits. In this linear scaling approach, the heat load due to the pulse generator itself is no doubt reduced, however, the heat load on other RF components (like the attenuators and filters along the flux-drive line of the generator in Extended Data Fig. 1) is still a problem. Therefore, I do not think the proposed pulse generator is a game changer.

OK, let's see what other approaches can offer.

CMOS and photonic links: Well, this also scales linearly with the number of qubits. However, the multiplexing capability in these approaches can noticeably reduce the required room-temperature RF equipment/components. The heat load can be an issue, however, there are numerous proposals to mitigate the problem [such as Sanskriti Joshi et al., Journal of Lightwave Technology 1, 166-175 (2024)].

Josephson junction lasers: This approach can potentially enable a cascading operation thanks to its large output power, i.e., using one primary on-chip generator to synchronize numerous secondary on-chip generator and then using the secondary generator for qubit control and read-out. Also

significantly reduce the required room-temperature RF equipment/components. As for lacking of waveform controllability, this issue may be overcome by replacing the single Josephson junction with a SQUID and integrating the generator with other on-chip superconducting components.

SFQ device: Still, this approach scales linearly. However, the SFQ device can enable fast gate operation. The quasi-particle poisoning can be a problem in long run, however, the signal quality is sufficient for control and read-out for state-of-the-art superconducting qubit as demonstrated in Ref. [18].

Minor issues:

1. In Fig. 2h, the author demonstrated the frequency tunability. It seems that the authors focused on the sweet spot in the rest part of the manuscript, how about the signal quality at other frequency? Does flux noise affect the signal quality noticeably?

2. I cannot not find any comments on the pulse duration of the generated pulse.

3. The authors demonstrate the qubit read-out, how about the qubit drive? Does the proposed pulse generator fulfil the requirements of fast gate operation?

Reviewer #2 (Remarks to the Author):

In the manuscript titled "A cryogenic on-chip microwave pulse generator..." authored by Zenghui Bao et al., the authors described a technique for generating microwave signals in cryostats for the readout of superconducting qubits at a frequency of few GHz. The technique utilizes a superconducting half-wave resonator with a SQUID loop at the node for the fundamental mode. By suddenly pulsing on the flux of the SQUID, the resonator can produce coherent microwave photons. The mechanism behind has been well understood and the authors provided added value by performing extensive characterizations and the showcase of reading-out a real qubit with it. Overall, I think Nature Communications is good venue to present this work, with only a few questions that must be answered by the authors:

1. The authors claim that the presented method would solve the scaling bottleneck. I see that one generator only generate signals at a particular frequency and a second one is even needed in order to have some tunability. Therefore, each qubit will need 1-2 such generators at cryogenic temperature and thus 1-2 channels coming from room temperature. I don't see how this can ease the scaling challenge. Please elaborate on this.
2. With two generators superimposed, I understand that pulse shaping can be implemented to some extent. However, I still expect that such shaping method is nowhere comparable to using commercial AWGs at room temperature or AWGs based on cryo-CMOS. They authors should comment on this. For example, if I want a Gaussian-shaped signal, how would that be implemented exactly and what's the limit of the bandwidth in such shaping?
3. Throughout the paper, power is plotted using arbitrary units. Are the scalings in different plots identical? How to convert the arbitrary units to real power in dBm?
4. Is the linewidth of the generated signal a function of power?
5. How is the qubit interconnected with the generator? What cable is used here?

Comments about “A cryogenic on-chip microwave pulse generator for large-scale superconducting 2 quantum computing”

To the authors:

The manuscript by Bao *et al* describes the experimental realization of a microwave signal generator using a SQUID based resonator, flux driven by digital pulses. Using only pulses with a spectral content from dc to about 1GHz, they demonstrate the generation of signals at about 5GHz, and the technique seems compatible with any frequency, at least in the 3-12GHz band typical for superconducting quantum circuits. They demonstrate basic control over amplitude, frequency, and phase. In addition, they use this system to drive the readout cavity of a 3D transmon and obtain standard readout fidelity. As far as I know the work is novel and might have potential for future applications. However, the discussion is at times unclear, and a lot of open questions remain. Provided that the authors clarify these questions and strengthen the manuscript, I believe this work would meet the requirement for publication in Nature Communication.

In particular, the main selling point is the potential scalability of this approach, with the author imagining a QPU with a large number of qubits addressed via low-bandwidth, low heat load cable (i.e twisted pairs). However, this work, in its current shape, does not fully support the feasibility of such approach. The major bottleneck for scaling signal delivery remains the qubit drives (qubit readout uses weaker signals and is easier to multiplex). It is very unclear how the authors address this issue here.

1. Can this technique deliver enough power to drive rabi oscillations at a reasonable rate, like 10MHz? Without having to take the data, one could simply do the back of the envelope calculation, $\Omega_R = 2\sqrt{\dot{n}\Gamma_{ext}}$, assuming a reasonable coupling rate Γ_{ext} between a drive line and a qubit (say $\Gamma_{ext} = 1ms^{-1}$), and \dot{n} the photon flux exiting the mw-source.
2. Throughout the paper, the authors alternate between voltage, photon number, power and power spectral density. In addition, the authors use various reference planes and sometime the calibrations are unclear. For example, Fig.2d is a voltage trace of the emitted signal (as digitized at room temperature?) but Fig.2g is in photon number (calibrated with the ac-stark shift?). In Fig.2g, using units of photons can be confusing as it is a unit of energy, which assumes that the signal was integrated over some amount of time, but which time? Or is it converted using the CW calibration and taking the peak value of the voltage of a trace like Fig.2d? Using photon flux unit (photon/second, i.e. $P/\hbar\omega$) might be useful to remove the contribution of the resonator linewidth

3. Can you expand on how the output signal varies with the resonator linewidth? Intuitively, the energy stored in the source being constant, faster mw-source decay means a higher power output for a shorter amount of time?
4. How would you go about generating more complex waveforms like DRAG? Or even just a gaussian pulse? Are you limited to decaying exponentials set by the mw-source linewidth?
5. From a naïve point of view, the generation of the signal relies on decay rate of the mw-source to the feedline. This works fine when driving the readout resonator, as a circulator ensure that the feedline looks dissipative. However, if one envisions driving directly a qubit drive line, then there is no dissipation, and the mw-source linewidth would go to zero. One would end up in a situation akin to resonant coupling with coherent exchange between the mw-source and the qubit. How would this work? Of course, a simple solution could be to use a circulator in between, but that's not scalable, or an attenuator, but one might be starved for power at this point.
6. Most of the work is done delivering the digital signal via a coax cable, with little to no benefit in terms of heat load compared to traditional signal delivery methods. Only a small part of the SI is dedicated to the use of twisted pairs, with an unclear conclusion. Without back of the envelope calculation for Rabi rate, it is hard for the reader to judge if this could actually drive a qubit using twisted pair (which is the main selling point of the article).

Other comments:

- Line 44: "qubits that *is* integrated"
- Line 144: what is the linewidth of the resonator?
- Line 156-161: explain how this is calibrated
- Line 163: "increasing the change rate of flux step" (sic) is not fully possible without going to coax cables, which defeat the purposes of this technique.
- Extended Data Fig.1: it looks like there are mw-switches before and after the mw-source, as well as at room temperature? Please label accordingly.
- Extended Data Fig.4: it might be useful to remind the reader that CW microwave emission is generated with digital pulse train.
- Extended Data Fig.5: how is the flux calibrated? Is it just using the "dc-voltage to flux calibration" and applying it to the digital voltage setpoints? In this case, it is not surprising that pulse distortion plays a big role. There is probably a way to calibrate the actual flux waveform by monitoring the cavity frequency at the function of time.
- SI, after (S7): is the approximation valid considering the large flux change in this system?

Reply to the Report of Referee A

In the manuscript entitled “A cryogenic on-chip microwave pulse generator for large-scale superconducting quantum computing”, Zenghui Bao et al., reported the realization of a compact on-chip pulse generator, different from the more well-established SFQ device, and demonstrated that the pulse generator can potentially meet the requirement of driving superconducting qubit.

It is an interesting technical paper. The topic, mitigating the scalability problem in a large-scale quantum computing system, is certainly of broad interest. However, I am not convinced that the work presents the level of novelty that warrants the publication in Nature Communications.

We sincerely thank the referee for taking the time to review this work and providing us the valuable comments. We are happy to see that the referee considers this work “interesting” and the research topic “of broad interest”. At the same time, we understand that the referee has reservations about the significance of our result and would like to take this opportunity to plead our case by extended discussion and outlook on addressing the scaling issue based on the demonstrated pulse generator. Below we carefully address the questions and concerns raised by the referee and list the corresponding revisions. We sincerely hope that with these changes and our point-by-point response below, the referee now will support the publication of our work in Nature Communications.

Major issue:

The major issue is, in my view, what is the advantage of this work? After all, the number of required pulse generator scales linearly with the number of qubits. In this linear scaling approach, the heat load due to the pulse generator itself is no doubt reduced, however, the heat load on other RF components (like the attenuators and filters along the flux-drive line of the generator in Extended Data Fig. 1) is still a problem. Therefore, I do not think the proposed pulse generator is a game changer.

We agree with the referee that we could have done a better job in emphasizing more clearly the advantages of the current pulse generator. We note that the purpose of this work is not to show a complete solution for large-scale superconducting quantum computers, which is too ambitious for a single paper. Instead, we aim to demonstrate this “novel” cryogenic pulse generator, which “might have potential for future applications” for scalable large-scale quantum computers, as commented by Referee C. In the revised manuscript, we have discussed some immediate advantages provided by the pulse generator to large-scale quantum computers, and also some technological advancements that are required to realize a true scalable quantum computer but are not provided by this work. Accordingly, the advantage of this pulse generator can be briefly summarized as

- 1, providing the possibility to improve the compactness of quantum chip packaging, and thus the integration density and connection reliability,
- 2, improve the economic feasibility compared with the expensive microwave devices,

3, reduce the heat load related to thermal noise attenuation.

We fully agree with the referee that using coax cables and attenuations to drive the pulse generator does not provide any advantage to the scaling problem. In the current work, the coax cables and attenuations are used to perform a full characterization of the pulse generator. However, such a tens of GHz channel is not necessary to operate the generator. As commented by Referee C, the pulse generator uses “only pulses with a spectral content from DC to about 1GHz”, and generates “signals at about 5GHz”, which “seems compatible with any frequency, at least in the 3-12GHz band typical for superconducting quantum circuits”. In the experiment, we show that microwave emission is possible even driven through a twisted pair of wires with a bandwidth of lower than 5 MHz (as shown in the newly added Fig. S5b). Although the emission power is reduced, we note that by optimizing the pitch size of the twisted pairs, the bandwidth can be greatly improved, and thus the emission power.

At the same time, we believe that a complete solution to address all the bottlenecks faced by today’s superconducting quantum computer also relies on the development of other technologies to optimize the control of the pulse generator and to realize monolithic integration of all necessary components, including optimized design of the twisted pairs of wires and cryogenic digital signal generator, etc.

We do believe that a single technological advancement is not enough to fully address the scaling problems for superconducting quantum computers. Considering that the pulse generator can be actuated with a digital signal, we expect that cryo-CMOS technology and SFQ electronics that provide either scalable signal edge or fast and accurate voltage pulses can be used to drive the pulse generator demonstrated here, which would breed a scalable superconducting quantum computer.

Revision: we have rewritten the last section of the manuscript

Discussion and outlook. In this work, we have realized a coherent on-chip cryogenic microwave pulse generator with negligible heat load and simple control, which provides distinct benefits for large-scale superconducting quantum computing. Benefiting from the small footprint and the compatibility with superconducting qubits, the pulse generator can be monolithically integrated with superconducting quantum circuits without using macroscale wiring harnesses and connectors, which will greatly improve the density and reliability of the input-output packaging [7–9]. Replacing the expensive microwave electronics and devices with chip-scale integrated circuits will also provide great economic feasibility to large-scale superconducting quantum computers. Moreover, compared with the conventional method, microwave pulses directly generated at the millikelvin temperatures can be sent to the qubits without the need for attenuation to suppress blackbody radiation from higher temperatures, which lessens a considerable amount of heat load at the millikelvin region, especially when operating a large number of qubits [15, 23, 43].

In the current prototypical demonstration, coaxial cables are used to deliver the driver signal of the pulse generator. We note that given the small bandwidth requirement, the pulse generator can be actuated even through superconducting twisted pairs of wires, for which the passive heat load is several orders of magnitude smaller than that of the coaxial cables [23]. Moreover,

deep cryogenic devices that are capable of generating the signal edges or voltage pulses required for the operation of the microwave source have been demonstrated before [10, 18]. Therefore, we anticipate that it may be possible to compactly integrate the microwave pulse generators demonstrated here and their controlling circuitries directly in the cryogenic environment, which functions as the control units for the quantum processor and can be controlled with digital instructions provided by a small number of wires connected to the room temperature electronics, as illustrated in Fig. 1b. Such a tight integration would lift a major roadblock toward large-scale superconducting quantum computation [7].

OK, let's see what other approaches can offer.

CMOS and photonic links: Well, this also scales linearly with the number of qubits. However, the multiplexing capability in these approaches can noticeably reduce the required room-temperature RF equipment/components. The heat load can be an issue, however, there are numerous proposals to mitigate the problem [such as Sanskriti Joshi et al., *Journal of Lightwave Technology* 1, 166-175 (2024)].

We thank the referee for bringing us this nice reference, where the limitations of both cryogenic CMOS and photonic links are discussed in great detail. This reference concludes that it is possible to scale to about 1000s qubits with those methods. Further scaling would be limited by the use of coaxial cables from the cold plate to the deep cryogenic temperature. It is also worth noting that connecting those coax cables to the superconducting chip with high yield is not a trivial task at all.

We agree with the referee that using cryo-CMOS and photonic links can greatly reduce the cost of RF equipment/components at room temperature. The pulse generator has a similar advantage in this aspect since it only requires a digital-like signal for operation, which is much cheaper and more scalable than expensive analog RF equipment/components.

Since both the cryo-CMOS and the photonic devices generate significant active heat load, they cannot be directly integrated with superconducting quantum circuits, which means coax wiring between those devices and the quantum chip is inevitable. However, for large-scale quantum computers containing millions of qubits, such macroscopic wirings and connections are not favored, if not impossible. As a comparison, the pulse generator demonstrated here provides the possibility of monolithic integration with the quantum chip, which would allow a larger-scale integration.

Revision: We have added a new reference

[15] S. Joshi and S. Moazeni, Scaling up superconducting quantum computers with cryogenic rf-photonics, *Journal of Lightwave Technology* 42, 166 (2024).

Josephson junction lasers: This approach can potentially enable a cascading operation thanks to its large output power, i.e., using one primary on-chip generator to synchronize numerous secondary on-chip generator and then using the secondary generator for qubit control and read-out. Also

significantly reduce the required room-temperature RF equipment/components. As for lacking of waveform controllability, this issue may be overcome by replacing the single Josephson junction with a SQUID and integrating the generator with other on-chip superconducting components.

We do agree with the referee that the Josephson junction laser provides a great choice as a continuous-wave microwave source. But using such devices for qubit readout and drive is still challenging. In addition to the lack of waveform controllability mentioned by the referee, the Josephson junction lasers cannot provide a well-controlled phase due to their intrinsic random initial phase. As a comparison, the pulse generator discussed in the manuscript can output microwave signals with a well-controlled phase, which is necessary for coherent control of superconducting qubits.

SFQ device: Still, this approach scales linearly. However, the SFQ device can enable fast gate operation. The quasi-particle poisoning can be a problem in long run, however, the signal quality is sufficient for control and read-out for state-of-the-art superconducting qubit as demonstrated in Ref. [18].

Yes, we agree with the referee that SFQ devices provide a promising approach to realizing large-scale superconducting quantum circuits, especially considering the recent advance in realizing high-fidelity qubit operation with SFQ electronics and mitigating the influence of quasi-particle poisoning. At the same time, we argue that scaling the SFQ electronics for millions of qubits is not straightforward since the cabling problem also exists in SFQ electronics. Besides, one has to deal with the signal picking-up at the millikelvin zone and the related heat load problems. It is also worth noting that compared with the SFQ circuits, the pulse generator demonstrated here is of simpler structure and better feasibility for integrating with superconducting qubits.

We believe that a single technology cannot fully address the scaling problem. In the experiment, we report that a steeper slope of the trigger signal can significantly increase the generated photon number. SFQ electronics is thus a great choice as a cryogenic controller of the pulse generator demonstrated here, which could reduce the bandwidth of SFQ drive signal to hundreds or even tens of kHz.

Minor issues:

1. In Fig. 2h, the author demonstrated the frequency tunability. It seems that the authors focused on the sweet spot in the rest part of the manuscript, how about the signal quality at other frequency? Does flux noise affect the signal quality noticeably?

Thanks for the nice comment. We note that due to the existence of subgap resistance of the SQUID, the internal loss rate of the resonator increases when it is tuned away from the sweet spot. Correspondingly, the output photon number decreases, as shown in the Extended Data Fig. 3. Other than that, we did not observe noticeable differences in terms of signal quality. For example, in the qubit readout experiment, we scan the generator frequency in about 20MHz range (Fig. 4b). In the qubit state single-shot measurement, the frequency of the generator is shifted by about 30MHz from the sweep spot. In those measurements, we did not observe the influence of flux noise on the signal quality.

2. I cannot not find any comments on the pulse duration of the generated pulse.

Thanks for the comment. The pulse duration is determined by the out-coupling rate of the cavity, as shown in the Extended Data Fig. 2 and discussed in lines 146 to 148 of the main text.

3. The authors demonstrate the qubit read-out, how about the qubit drive? Does the proposed pulse generator fulfil the requirements of fast gate operation?

Thanks for the comment. We agree with the referee that we did not provide enough details on qubit drive in the previous manuscript. In the revised manuscript, we have added more discussion on using the pulse generator for qubit drive.

Briefly, we evaluate two kinds of qubit drive schemes based on the current pulse generator. We conclude that for the conventional drive scheme, stronger output power is required to have a fast enough Rabi rate. For the so-called ‘energy saving gate’ scheme, the current pulse generator can support a fast enough Rabi rate.

For large-scale quantum computing with millions of qubits, the qubit drive signal in the conventional approach would generate a significant heat load that cannot be handled by the dilution fridge (*J. Ikonen, et. al., npj Quantum Information 3 (2017)*). In this sense, it is necessary to implement the ‘energy saving gate’, with which much smaller photons are required to achieve the same Rabi rate. We have added more detailed discussions in the revised manuscript.

Revision:

1, in the main text, we have added a new qubit drive section.

Qubit drive. We consider two approaches to perform the single qubit operation with the cryogenic microwave source and theoretically estimate the possibly achieved single qubit gate speed, respectively, as shown in the Extended data Fig. 6. Similar to the conventional method, we can connect the output of the signal source to the qubit through an open waveguide. In this case, the Rabi rate that can be achieved is on the order of 0.26 MHz, with the maximum output power obtained so far and reasonable coupling strength between the waveguide and the qubit, which is still too slow for practical superconducting quantum computing. Alternatively, we can directly connect the resonator of the signal source to the qubit and tune them on resonance when performing the qubit gate operation. With a similar coupling strength and photon numbers, the estimated Rabi rate is about 30 MHz, which is much faster than that of the conventional method and agrees well with the previous reports [42, 43]. We note that such an ‘energy-saving gate’ is preferred for large-scale quantum computing [43].

A stronger output power is preferred to realize fast and high-fidelity qubit state manipulation. A direct approach is to superimpose multiple microwave pulses to harvest more microwave photons with either multiple signal sources or using a series of flux steps/overshoots as drive, as demonstrated in Fig. 3b, c and d. Additionally, from our simulation results, stronger emission of a single microwave pulse can be obtained by increasing the flux change rate and fine-tuning the symmetry of the resistances of the Josephson junctions in the SQUID. A detailed discussion can be found in the Supplementary Information.

2, Correspondingly, in the METHODS, we have added a new section.

Qubit drive estimation. We consider two approaches to perform a single qubit gate with the microwave source, and estimate the possibly achieved single qubit Rabi rate.

The first approach is similar to the conventional qubit drive scheme, for which the output of the signal source is coupled to the qubit via an open waveguide. Taking the coupling strength between the waveguide and the qubit as Γ_{ext} , the Rabi rate Ω can be estimated as

$$\Omega = 2\sqrt{\dot{n}_d \Gamma_{ext}}$$

where \dot{n} is the photon flux in the waveguide. Considering the energy consumption during the single qubit π rotation, the Rabi rate can be expressed as the function of total input photon number n in the waveguide

$$\Omega = \frac{4}{\pi} n \Gamma_{ext}$$

Here, we take a reasonable coupling strength $\Gamma_{ext} = 1000$ Hz for calculation, corresponding to a qubit lifetime limitation 1 ms, and the result is shown in the Extended data Fig. 6 with the purple line. According to the maximum output power obtained with the cryogenic microwave source, the achievable Rabi rate is about 0.26 MHz, which is too slow compared with the practical qubit gate speed implying an insufficient output power.

In the second approach, the resonator of the signal source is directly connected to the qubit through a capacitive coupling. The coupling capacitance is chosen to be the same as that between the waveguide and the qubit in the first approach. The resonator and the qubit are tuned on resonance when performing the gate operation and tuned far off-resonance when idle. To simulate the single qubit gate, we consider the Jaynes-Cummings Hamiltonian

$$\hat{H} = \frac{\omega_q}{2} \hat{\sigma}_z + \omega_r \hat{a}^\dagger \hat{a} + g(\hat{a}^\dagger \hat{\sigma}_- + \hat{a} \hat{\sigma}_+),$$

where \hat{a} (\hat{a}^\dagger) is the annihilation (creation) operator of the resonator mode, and $\hat{\sigma}$ is the Pauli operator of the qubit mode; ω_q and ω_r are the frequency of the qubit and resonator of the cryogenic signal source, respectively; g is the coupling strength between the qubit and the resonator. Since the coupling strengths Γ_{ext} and g in the two approaches are both determined by the coupling capacitance C_c , it is intuitive to compare the Rabi rates of the two approaches under the same coupling capacitance. According to a simple capacitive coupling model, the relations between the two coupling strengths and the coupling capacitance can be expressed as

$$C_c = \sqrt{\frac{C_q \Gamma_{ext}}{\omega_q^2 Z_0}},$$

$$g = \frac{C_c}{2C_q C_r} \sqrt{\frac{1}{Z_r Z_q}}$$

where C_q and C_r are the capacitance of the qubit and the resonator; Z_0 , Z_q , and Z_r are the characteristic impedances of the open waveguide, the qubit, and the resonator. With a common qubit parameter setting $\omega_q = 6$ GHz, $E_c = 220$ MHz, the coupling strength $\Gamma_{ext} = 1000$ corresponds to the coupling capacitance $C_c = 35$ aF. Therefore, the coupling strength in the second approach can be determined as $g = 0.49$ MHz. By considering the coherent states of different photon numbers in the resonator, the simulation results of the related Rabi rate are shown in Extended data Fig. 6 with the pink line. In this approach, the Rabi rate is much faster than the conventional method with the same photon number. The pink triangle and square indicate the Rabi rates 30.8 MHz and 8.23 MHz corresponding to the maximum photon number experimentally generated in the resonator 1017 and 71 when the flux drive signal of the cryogenic microwave source is delivered through the coaxial cables or twisted-pair wires. By optimizing the twist pitch length of the pair of wires and thus their bandwidth [31], the

maximum achieved photon number is estimated to be 662, leading to the optimized Rabi rate of 24.9 MHz, as shown with the pink diamond in the Extended data Fig. 6.

Extended Data Fig. 6. Qubit drive estimation with the cryogenic microwave source. The single qubit Rabi rate is theoretically estimated as a function of the emission photon number with the conventional method (purple line) or the energy-saving gate approach (pink line). In the conventional approach, the output of the signal source is connected to the qubit through an open waveguide as the drive line, where the coupling rate between the qubit and the drive line is assumed to be $\Gamma_{\text{ext}} = 1000$ Hz for theoretical estimation. In this case, the Rabi rate achieved with the maximum emission output power using coaxial cables is on the order of 0.26 MHz marked by the purple circle. For the energy-saving gate approach, the resonator of the signal source is directly connected to the qubit with a coupling capacitance the same as the conventional method [42, 43]. Thus, the single-qubit operation is realized by resonant Rabi oscillation between the qubit and resonator. The pink triangle and square indicate the Rabi rates 30.8 MHz and 8.23 MHz corresponding to the maximum photon number experimentally generated in the resonator 1017 and 71 when the flux drive signal of the signal source is delivered through the coaxial cables or twisted-pair wires. By optimizing the twist pitch length of the pair of wires and thus their bandwidth [31], the maximum achieved photon number is estimated to be 662 with the Rabi rate 24.9 MHz shown with the pink diamond. The schematic diagrams of the two approaches are shown below the corresponding lines, respectively.

Reply to the Report of Referee B

In the manuscript titled “A cryogenic on-chip microwave pulse generator...” authored by Zenghui Bao et al., the authors described a technique for generating microwave signals in cryostats for the readout of superconducting qubits at a frequency of few GHz. The technique utilizes a superconducting half-wave resonator with a SQUID loop at the node for the fundamental mode. By suddenly pulsing on the flux of the SQUID, the resonator can produce coherent microwave photons. The mechanism behind has been well understood and the authors provided added value by performing extensive characterizations and the showcase of reading-out a real qubit with it. Overall, I think Nature Communications is good venue to present this work, with only a few questions that must be answered by the authors:

We thank the referee for taking the time to review our manuscript and providing us the valuable and insightful comments, which have helped us greatly in improving the quality of our manuscript. We are very grateful for the positive assessment of our work and for supporting its publication in Nature Communications. Below we will try to address the referee’s concerns and list the corresponding revisions.

1, The authors claim that the presented method would solve the scaling bottleneck. I see that one generator only generate signals at a particular frequency and a second one is even needed in order to have some tunability. Therefore, each qubit will need 1-2 such generators at cryogenic temperature and thus 1-2 channels coming from room temperature. I don’t see how this can ease the scaling challenge. Please elaborate on this.

We thank the referee for the comment. We agree with the referee that we could have done a better job of explaining more about how the current pulse generator helps in addressing the scaling challenge of superconducting quantum computing. In the revised manuscript, we have added a more detailed discussion of the benefits brought by the pulse generator.

We fully agree with the referee that using coax cables to operate the pulse generator does not ease the scaling problem. In the manuscript, we show that digital-like signals delivered by ribbon cables can be used to drive the pulse generator. Benefiting from its low passive thermal load, using ribbon cables to deliver the drive signal would allow two orders of magnitudes more channels integrated into the dilution fridge compared with coax cables. Meanwhile, deep-cryogenic devices that are capable of generating digital signals in a scalable approach have been demonstrated before (e.g., *S. Pauka, et. al., Nature Electronics 4, 64 (2021).*). Therefore, we envision that the demonstrated pulse generator and its controlling circuitries can be directly integrated into the cryogenic environment, which functions as the control units for the quantum chips and is controlled with digital instructions provided by a small number of wires connected to the room temperature electronics.

In the current work, we did not manage to demonstrate a full spectrum of technologies to ultimately solve the scaling of superconducting quantum computers. The main purpose is to

introduce this cryogenic pulse generator, which to our knowledge is the first time to report such a photon generation process in the deep cryogenic environment. Some immediate advantages that the generator brings to the scaling problem are discussed in the revised manuscript and can be summarized as

- 1, providing the possibility to improve the compactness of quantum chip packaging, and thus the integration density and connection reliability,
- 2, improve the economic feasibility compared with the expensive microwave devices,
- 3, reduce the heat load related to thermal noise attenuation.

Revision: we have rewritten the last section of the manuscript

Discussion and outlook. In this work, we have realized a coherent on-chip cryogenic microwave pulse generator with negligible heat load and simple control, which provides distinct benefits for large-scale superconducting quantum computing. Benefiting from the small footprint and the compatibility with superconducting qubits, the pulse generator can be monolithically integrated with superconducting quantum circuits without using macroscale wiring harnesses and connectors, which will greatly improve the density and reliability of the input-output packaging [7–9]. Replacing the expensive microwave electronics and devices with chip-scale integrated circuits will also provide great economic feasibility to large-scale superconducting quantum computers. Moreover, compared with the conventional method, microwave pulses directly generated at the millikelvin temperatures can be sent to the qubits without the need for attenuation to suppress blackbody radiation from higher temperatures, which lessens a considerable amount of heat load at the millikelvin region, especially when operating a large number of qubits [15, 23, 43].

In the current prototypical demonstration, coaxial cables are used to deliver the driver signal of the pulse generator. We note that given the small bandwidth requirement, the pulse generator can be actuated even through superconducting twisted pairs of wires, for which the passive heat load is several orders of magnitude smaller than that of the coaxial cables [23]. Moreover, deep cryogenic devices that are capable of generating the signal edges or voltage pulses required for the operation of the microwave source have been demonstrated before [10, 18]. Therefore, we anticipate that it may be possible to compactly integrate the microwave pulse generators demonstrated here and their controlling circuitries directly in the cryogenic environment, which functions as the control units for the quantum processor and can be controlled with digital instructions provided by a small number of wires connected to the room temperature electronics, as illustrated in Fig. 1b. Such a tight integration would lift a major roadblock toward large-scale superconducting quantum computation [7].

2, With two generators superimposed, I understand that pulse shaping can be implemented to some extent. However, I still expect that such shaping method is nowhere comparable to using commercial AWGs at room temperature or AWGs based on cryo-CMOS. They authors should comment on this. For example, if I want a Gaussian-shaped signal, how would that be implemented exactly and what's the limit of the bandwidth in such shaping?

Thanks for the nice comment. We have to admit that with only the pulse generator, we cannot compete with a commercial AWG in terms of generating arbitrary waveform. This is indeed a flaw of this device. If an arbitrary waveform is necessary, one could consider to introduce a tunable out-coupling of the pulse generator to the external circuit. Besides, we note that for qubits with large anharmonicity, the influence of pulse shape can be mitigated.

Revision: line 148 to 151, we have added

...We note that tuning the envelope of the microwave pulse to realize the desired waveform is possible by introducing a tunable out-coupling of the signal source to the external circuits [29,30]. ...

3, Throughout the paper, power is plotted using arbitrary units. Are the scalings in different plots identical? How to convert the arbitrary units to real power in dBm?

Thanks for the nice suggestion. We agree with the referee that the use of power units is a bit confusing. With the arbitrary units, we just want to show the evolution of the emitted power versus some parameters like start/stop flux. To avoid possible misunderstandings, in the revised manuscript, we replaced most of the energy units with photon numbers, including Fig.2e, Fig.2g, Fig.3b, Extended Data Fig. 3, and Extended Data Fig. 5. At the same time, we kept some of the time traces still in the voltage unit to keep an intuitive picture of the photon emission process (Fig.2d, Fig.3d, and Extended Data Fig. 2). We have also added more descriptions in the corresponding captions.

Revision: The unit of the vertical axis for Fig.2e, Fig.2g, Fig.3b, Extended Data Fig. 3, and Extended Data Fig. 5 are changed to photon number.

4, Is the linewidth of the generated signal a function of power?

Thanks for the nice question. Considering that the circuits contain non-linear elements, the self-Kerr effect could play a role during the signal generation process. However, in the experimental results, we did not find frequency and linewidth dependence on the emission power. Considering the small junction inductance, this may indicate that the generated power is still not enough to introduce the observable Kerr effect.

5, How is the qubit interconnected with the generator? What cable is used here?

Thanks for the comment. In the experiment, the qubit is connected to the generator with coaxial cables. This is indeed an insightful comment since we also consider that bulky coaxial cable is not a good choice for large-scale integration. However, as a proof-of-principle experiment, we did not manage to use a tightly integrated design.

In principle, for both qubit readout and control, the generator can be monolithically integrated with the qubit. For qubit state readout, such an integration is intuitive. For qubit state control, we plan to directly couple the generator with the qubit, using the so-called ‘energy-saving gate’

instead of the conventional method. We have added a new section to discuss the qubit drive schemes in the revised manuscript.

Reply to the Report of Referee C

The manuscript by Bao et al describes the experimental realization of a microwave signal generator using a SQUID based resonator, flux driven by digital pulses. Using only pulses with a spectral content from dc to about 1GHz, they demonstrate the generation of signals at about 5GHz, and the technique seems compatible with any frequency, at least in the 3-12GHz band typical for superconducting quantum circuits. They demonstrate basic control over amplitude, frequency, and phase. In addition, they use this system to drive the readout cavity of a 3D transmon and obtain standard readout fidelity. As far as I know the work is novel and might have potential for future applications. However, the discussion is at times unclear, and a lot of open questions remain. Provided that the authors clarify these questions and strengthen the manuscript, I believe this work would meet the requirement for publication in Nature Communication.

We sincerely thank the referee for the thorough review of our manuscript and for providing us the valuable and insightful comments, which indeed helped us a lot in improving the manuscript and reconsidering this work. Below we tried to address the referee's comments point by point and we hope that the reply and revision can resolve the concerns on this work.

In particular, the main selling point is the potential scalability of this approach, with the author imagining a QPU with a large number of qubits addressed via low-bandwidth, low heat load cable (i.e twisted pairs). However, this work, in its current shape, does not fully support the feasibility of such approach. The major bottleneck for scaling signal delivery remains the qubit drives (qubit readout uses weaker signals and is easier to multiplex). It is very unclear how the authors address this issue here.

We thank the referee for pointing out the weakness of our manuscript. In the revised manuscript, we have added more discussion about using the pulse generator for qubit drive. We have also made substantial revisions to express more clearly our strength in optimizing the scalability of SC quantum computers.

In the current manuscript, we did not manage to demonstrate the full spectrum of technology to scale up the SC quantum computer. This work aims to report the compact cryogenic microwave pulse generator and demonstrate its potential in qubit readout and drive, which we believe is the major roadblock toward truly scalable SC quantum computers. Using ribbon cable can mitigate the heat load problem to some extent, but it is still not a truly scalable architecture. Ideally, a truly scalable quantum computer can be accessed from room temperature via limited digital IO channels. In this sense, the pulse generator greatly reduces the complexity of the controlling electronics. For example, it has been demonstrated that deep-cryogenic digital signal generators can be realized with cryo-CMOS or SFQ technologies (e.g., *S. Pauka, et. al., Nature Electronics 4, 64 (2021).* and *L. Howe, et. al., PRX Quantum 3, 010350 (2022)*). We have also added related discussions in the revised manuscript.

Revision: we have rewritten the last section of the manuscript

Discussion and outlook. In this work, we have realized a coherent on-chip cryogenic microwave pulse generator with negligible heat load and simple control, which provides distinct benefits for large-scale superconducting quantum computing. Benefiting from the small footprint and the compatibility with superconducting qubits, the pulse generator can be monolithically integrated with superconducting quantum circuits without using macroscale wiring harnesses and connectors, which will greatly improve the density and reliability of the input-output packaging [7–9]. Replacing the expensive microwave electronics and devices with chip-scale integrated circuits will also provide great economic feasibility to large-scale superconducting quantum computers. Moreover, compared with the conventional method, microwave pulses directly generated at the millikelvin temperatures can be sent to the qubits without the need for attenuation to suppress blackbody radiation from higher temperatures, which lessens a considerable amount of heat load at the millikelvin region, especially when operating a large number of qubits [15, 23, 43].

In the current prototypical demonstration, coaxial cables are used to deliver the driver signal of the pulse generator. We note that given the small bandwidth requirement, the pulse generator can be actuated even through superconducting twisted pairs of wires, for which the passive heat load is several orders of magnitude smaller than that of the coaxial cables [23]. Moreover, deep cryogenic devices that are capable of generating the signal edges or voltage pulses required for the operation of the microwave source have been demonstrated before [10, 18]. Therefore, we anticipate that it may be possible to compactly integrate the microwave pulse generators demonstrated here and their controlling circuitries directly in the cryogenic environment, which functions as the control units for the quantum processor and can be controlled with digital instructions provided by a small number of wires connected to the room temperature electronics, as illustrated in Fig. 1b. Such a tight integration would lift a major roadblock toward large-scale superconducting quantum computation [7].

1. Can this technique deliver enough power to drive rabi oscillations at a reasonable rate, like 10MHz? Without having to take the data, one could simply do the back of the envelope calculation, $\Omega_R = 2\sqrt{n}\Gamma_{ext}$, assuming a reasonable coupling rate Γ_{ext} between a drive line and a qubit (say $\Gamma_{ext} = 1ms^{-1}$), and n the photon flux exiting the mw-source.

We thank the referee for the nice comment and suggestion. Evaluating how to use the pulse generator to drive the qubit is indeed an important problem, which is also one of our ongoing projects. In the revised manuscript, we estimate two different approaches for qubit drive based on the pulse generator.

One of the approaches is similar to the conventional method, where the pulse generator excites the qubit through a waveguide mode. In this case, the drive process can be estimated as the referee’s comment. For the largest photon flux currently achieved in our experiments, we can only reach a Rabi rate of about 0.26MHz, which is practically too slow. A larger output power is preferred for a larger Rabi rate. We have the feeling that the maximum achieved output power can be further improved since we consider this work only a proof-of-principle demonstration

of this kind of generator. There is still plenty of room for optimization both in terms of device design and fabrication.

In the second approach, the cavity in the pulse generator is directly coupled to the qubit. Since the cavity in the generator is frequency-tunable, we could align the cavity on resonance to the qubit when performing gate operation, and far detune it from the qubit in the idle time to preserve the coherence of the qubit. This configuration is known as an ‘energy-saving gate’ in previous works (*J. Ikonen, et. al., npj Quantum Information 3 (2017)*), with which a larger Rabi rate can be achieved for the same photon number. Our numerical simulation demonstrates that when the qubit-cavity coupling is the same as that of the qubit and the waveguide with $\Gamma_{ext} = 1\text{ms}^{-1}$, we can reach a Rabi rate of about 30.8MHz with only 1000 photons that were achieved in our current experiment. For a quantum computer with millions of qubits, an inefficient use of the qubit drive power would generate a significant heat load that cannot be handled by the dilution fridge. In this sense, the energy-saving gate is necessary for large-scale superconducting quantum computing.

Figure R1: **Illustrations for the qubit drive approaches.** **a**, In the conventional approach, the pulse generator is connected to the qubit through an open transmission line. One end of the transmission line is terminated to realize a continuous dissipation channel. **b**, The energy-saving gate scheme, where the resonator in the pulse generator is directly coupled to the qubit through the capacitor C_c .

Revision:

1, in the main text, we have added a new qubit drive section.

Qubit drive. We consider two approaches to perform the single qubit operation with the cryogenic microwave source and theoretically estimate the possibly achieved single qubit gate speed, respectively, as shown in the Extended data Fig. 6. Similar to the conventional method, we can connect the output of the signal source to the qubit through an open waveguide. In this case, the Rabi rate that can be achieved is on the order of 0.26 MHz, with the maximum output power obtained so far and reasonable coupling strength between the waveguide and the qubit, which is still too slow for practical superconducting quantum computing. Alternatively, we can directly connect the resonator of the signal source to the qubit and tune them on resonance when performing the qubit gate operation. With a similar coupling strength and photon numbers, the estimated Rabi rate is about 30 MHz, which is much faster than that of the conventional method and agrees well with the previous reports [42, 43]. We note that such an ‘energy-saving gate’ is preferred for large-scale quantum computing [43].

A stronger output power is preferred to realize fast and high-fidelity qubit state manipulation. A direct approach is to superimpose multiple microwave pulses to harvest more microwave photons with either multiple signal sources or using a series of flux steps/overshoots as drive, as demonstrated in Fig. 3b, c and d. Additionally, from our simulation results, stronger emission of a single microwave pulse can be obtained by increasing the flux change rate and fine-tuning the symmetry of the resistances of the Josephson junctions in the SQUID. A detailed discussion can be found in the Supplementary Information.

2, Correspondingly, in the METHODS, we have added a new section.

Qubit drive estimation. We consider two approaches to perform a single qubit gate with the microwave source, and estimate the possibly achieved single qubit Rabi rate.

The first approach is similar to the conventional qubit drive scheme, for which the output of the signal source is coupled to the qubit via an open waveguide. Taking the coupling strength between the waveguide and the qubit as Γ_{ext} , the Rabi rate Ω can be estimated as

$$\Omega = 2\sqrt{\dot{n}_d \Gamma_{ext}}$$

where \dot{n} is the photon flux in the waveguide. Considering the energy consumption during the single qubit π rotation, the Rabi rate can be expressed as the function of total input photon number n in the waveguide

$$\Omega = \frac{4}{\pi} n \Gamma_{ext}$$

Here, we take a reasonable coupling strength $\Gamma_{ext} = 1000$ Hz for calculation, corresponding to a qubit lifetime limitation 1 ms, and the result is shown in the Extended data Fig. 6 with the purple line. According to the maximum output power obtained with the cryogenic microwave source, the achievable Rabi rate is about 0.26 MHz, which is too slow compared with the practical qubit gate speed implying an insufficient output power.

In the second approach, the resonator of the signal source is directly connected to the qubit through a capacitive coupling. The coupling capacitance is chosen to be the same as that between the waveguide and the qubit in the first approach. The resonator and the qubit are tuned on resonance when performing the gate operation and tuned far off-resonance when idle. To simulate the single qubit gate, we consider the Jaynes-Cummings Hamiltonian

$$\hat{H} = \frac{\omega_q}{2} \hat{\sigma}_z + \omega_r \hat{a}^\dagger \hat{a} + g(\hat{a}^\dagger \hat{\sigma}_- + \hat{a} \hat{\sigma}_+),$$

where \hat{a} (\hat{a}^\dagger) is the annihilation (creation) operator of the resonator mode, and $\hat{\sigma}$ is the Pauli operator of the qubit mode; ω_q and ω_r are the frequency of the qubit and resonator of the cryogenic signal source, respectively; g is the coupling strength between the qubit and the resonator. Since the coupling strengths Γ_{ext} and g in the two approaches are both determined by the coupling capacitance C_c , it is intuitive to compare the Rabi rates of the two approaches under the same coupling capacitance. According to a simple capacitive coupling model, the relations between the two coupling strengths and the coupling capacitance can be expressed as

$$C_c = \sqrt{\frac{C_q \Gamma_{ext}}{\omega_q^2 Z_0}},$$

$$g = \frac{C_c}{2C_q C_r} \sqrt{\frac{1}{Z_r Z_q}}$$

where C_q and C_r are the capacitance of the qubit and the resonator; Z_0 , Z_q , and Z_r are the characteristic impedances of the open waveguide, the qubit, and the resonator. With a common qubit parameter setting $\omega_q = 6$ GHz, $E_c = 220$ MHz, the coupling strength $\Gamma_{\text{ext}} = 1000$ corresponds to the coupling capacitance $C_c = 35$ aF. Therefore, the coupling strength in the second approach can be determined as $g = 0.49$ MHz. By considering the coherent states of different photon numbers in the resonator, the simulation results of the related Rabi rate are shown in Extended data Fig. 6 with the pink line. In this approach, the Rabi rate is much faster than the conventional method with the same photon number. The pink triangle and square indicate the Rabi rates 30.8 MHz and 8.23 MHz corresponding to the maximum photon number experimentally generated in the resonator 1017 and 71 when the flux drive signal of the cryogenic microwave source is delivered through the coaxial cables or twisted-pair wires. By optimizing the twist pitch length of the pair of wires and thus their bandwidth [31], the maximum achieved photon number is estimated to be 662, leading to the optimized Rabi rate of 24.9 MHz, as shown with the pink diamond in the Extended data Fig. 6.

Extended Data Fig. 6. Qubit drive estimation with the cryogenic microwave source. The single qubit Rabi rate is theoretically estimated as a function of the emission photon number with the conventional method (purple line) or the energy-saving gate approach (pink line). In the conventional approach, the output of the signal source is connected to the qubit through an open waveguide as the drive line, where the coupling rate between the qubit and the drive line is assumed to be $\Gamma_{\text{ext}} = 1000$ Hz for theoretical estimation. In this case, the Rabi rate achieved with the maximum emission output power using coaxial cables is on the order of 0.26 MHz marked by the purple circle. For the energy-saving gate approach, the resonator of the signal source is directly connected to the qubit with a coupling capacitance the same as the conventional method [42, 43]. Thus, the single-qubit operation is realized by resonant Rabi oscillation between the qubit and resonator. The pink triangle and square indicate the Rabi rates 30.8 MHz and 8.23 MHz corresponding to the maximum photon number experimentally generated in the resonator 1017 and 71 when the flux drive signal of the signal source is delivered through the coaxial cables or twisted-pair wires. By optimizing the twist pitch length of the pair of wires and thus their bandwidth [31], the maximum achieved photon number is estimated to be 662 with the Rabi rate 24.9 MHz shown with the pink diamond. The schematic

diagrams of the two approaches are shown below the corresponding lines, respectively.

2. Throughout the paper, the authors alternate between voltage, photon number, power and power spectral density. In addition, the authors use various reference planes and sometime the calibrations are unclear. For example, Fig.2d is a voltage trace of the emitted signal (as digitized at room temperature?) but Fig.2g is in photon number (calibrated with the ac-stark shift?). In Fig.2g, using units of photons can be confusing as it is a unit of energy, which assumes that the signal was integrated over some amount of time, but which time? Or is it converted using the CW calibration and taking the peak value of the voltage of a trace like Fig.2d? Using photon flux unit (photon/second, i.e. $P/h\omega$) might be useful to remove the contribution of the resonator linewidth

Thanks for the nice suggestion. We agree with the referee that the use of power units in the previous manuscript is a bit confusing. In the revised manuscript, we remove most of the arbitrary units and replace the energy units with photon numbers, including Fig.2e, Fig.2g Fig.2h, Fig.3b, Extended Data Fig. 3, Extended Data Fig. 5, and Fig. S4. When the pulse generator is triggered with a single digital signal, it emits a microwave pulse with an exponential decay envelope (Fig.2d), for which the photon number is integrated over the pulse duration. When the pulse generator is triggered by a train of digital signals, it generates a continuous wave signal, for which we use power spectra density to show the spectrum information, including the formation of the frequency comb (Fig.3e), the narrow linewidth (Fig.3f), and the resolution-limited linewidth measurement (Extended Data Fig. 4).

At the same time, we keep some of the time traces still in the voltage unit to present an intuitive picture of the photon emission process (Fig.2d, Fig.3d, and Extended Data Fig. 2). We have also added more descriptions in the corresponding captions.

Revision:

1, The unit of the vertical axis for Fig.2e, Fig.2g, Fig.3b, Extended Data Fig. 3, and Extended Data Fig. 5 are changed to photon number. The captions of those figures are also modified accordingly.

2, In the caption of Fig. 3e, we have added ‘The power spectral densities have been normalized and offset for comparison purposes’.

3. Can you expand on how the output signal varies with the resonator linewidth? Intuitively, the energy stored in the source being constant, faster mw-source decay means a higher power output for a shorter amount of time?

Thanks for the suggestion. We would say yes. Even though we do not have a device with a tunable out-coupling rate, from the dependence on the internal loss rate, we could conclude that the total energy does not change as a function of cavity linewidth. At the same time, since the pulse leak time constant is determined by the linewidth, we could conclude that a larger out-coupling rate would yield higher output power.

Revision: line 181 to 185, we have added

... Meanwhile, by increasing the out-coupling rate of the signal source to the external circuits, the generated microwave photons can leak out faster, thus achieving stronger emission power. ...

4. How would you go about generating more complex waveforms like DRAG? Or even just a gaussian pulse? Are you limited to decaying exponentials set by the mw-source linewidth?

Thanks for the comment. The referee is right that for the simplest version of the pulse generator, the output pulse has an exponential envelope that is determined by the linewidth of the cavity in the generator. For the conventional qubit drive method and for superconducting qubits with small anharmonicity, this could be a potential problem. An additional pulse shaper may be required to mitigate undesired drive leakage. For qubits with large anharmonicity, such as fluxonium or C-shunt flux qubit, the lack of a pulse shaper might not be a problem.

However, we note that for current technology, an arbitrary waveform generation is too expensive for large-scale superconducting quantum computing, both economically and in terms of heat load. The current device that generates a microwave pulse with a well-defined phase is already a clear advantage.

Revision: line 148 to 151, we have added

... We note that tuning the envelope of the microwave pulse to realize the desired waveform is possible by introducing a tunable out-coupling of the signal source to the external circuits [29,30]. ...

5. From a naïve point of view, the generation of the signal relies on decay rate of the mwsource to the feedline. This works fine when driving the readout resonator, as a circulator ensure that the feedline looks dissipative. However, if one envisions driving directly a qubit drive line, then there is no dissipation, and the mw-source linewidth would go to zero. One would end up in a situation akin to resonant coupling with a coherent exchange between the mw-source and the qubit. How would this work? Of course, a simple solution could be to use a circulator in between, but that's not scalable, or an attenuator, but one might be starved for power at this point.

We thank the referee for this insightful comment. As also mentioned in our reply to the previous comment, we have considered two approaches to perform the qubit drive, as illustrated in Figure R1.

For the first approach, as also commented by the referee, we could drive the qubit with the conventional method with the pulse generator, that is, sending microwave pulses to the qubit through a continuous channel that is weakly coupled to the qubit. The continuous channel can be connected the pulse generator at one side, while terminated at the other side to form a continuous channel, as illustrated in Figure S1, for which we do not need the bulky circulator or attenuation in between the qubit and the generator to consume more power. In this case, one needs an on-chip attenuator/termination (*Jen-Hao Yeh, et. al. J. Appl. Phys. 121, 224501 (2017)*)

design to realize monolithic integration. As discussed in the revised manuscript and also in our reply to the previous comment, for the current situation the output power of the generator is not enough to perform a fast enough single qubit gate.

For the second approach, we directly couple the generator with the qubit. We tune the cavity and the qubit in resonance when performing the single qubit gate, and largely detune them during the idling time. According to previous works and also reproduced by our simulation results, such a gate scheme could realize the same Rabi rate but using much smaller energy than the conventional method, while still preserving a high enough gate fidelity.

6. Most of the work is done delivering the digital signal via a coax cable, with little to no benefit in terms of heat load compared to traditional signal delivery methods. Only a small part of the SI is dedicated to the use of twisted pairs, with an unclear conclusion. Without back of the envelope calculation for Rabi rate, it is hard for the reader to judge if this could actually drive a qubit using twisted pair (which is the main selling point of the article).

Thanks for the comment. We agree with the referee that in the old manuscript, our discussion on the use of twisted pairs for qubit manipulation is not enough. In the revised manuscript, we have add more details about the property of the used twisted pairs and estimated the maximum achievable Rabi rate for our current situation, which definitely can be improved by optimizing the twisted pairs of wires and the generator design.

Revision:

1, Lines 207 to 211, we have added

... be effectively generated, but with a lower output power, which is typically about 0.07 times that of the coaxial cables and can be improved by optimizing the design of the twisted pairs. Detailed results ...

2, We have added a new Fig. S5b to show the bandwidth measurement of the twisted pairs.

3, We have added a new Extended Data Fig. 6 to show the estimated Rabi rate when using the twisted pairs to deliver the flux drive signal.

Other comments:

- Line 44: "qubits that is integrated"

We appreciate the referee for catching this typo and have corrected it in the revised manuscript.

Revision: line 44

qubits that is integrated

- Line 144: what is the linewidth of the resonator?

The typical linewidth is about 0.5 MHz to 1MHz, which is mainly determined by the outcoupling rate to the transmission line.

- Line 156-161: explain how this is calibrated

Thanks for the suggestion. We have added some descriptions of photon number calibration in the main text.

Revision: line 138 to 141, we have added

... The microwave photon number is calibrated based on the measurement-induced dephasing of a superconducting qubit (Supplementary Information, section III). ...

- Line 163: “increasing the change rate of flux step” (sic) is not fully possible without going to coax cables, which defeat the purposes of this technique.

Thanks for the nice comment. With this context, we just want to describe one of the basic properties of this pulse generator. We agree with the referee that applying a steeper flux step is somehow impossible because of the limited bandwidth of the ribbon cables. The property could be important when using some cryogenic digital source (e.g., SFQ electronics) to drive the pulse generator, where the cabling problem can be mitigated by monolithic integration.

- Extended Data Fig.1: it looks like there are mw-switches before and after the mw-source, as well as at room temperature? Please label accordingly.

Thanks for the comment. We did not use microwave switches in the experiments. Those switch-like devices in the diagram are used to distinguish different input/output circuits. To avoid possible misleading, we have changed the plot and clarified it in the revised manuscript.

Revision:

Extended Data Fig. 1. **Schematic diagram of the experimental setup**.... The dashed boxes

enclosed area designate different circuit connections depending on the specific measurements. ...

- Extended Data Fig.4: it might be useful to remind the reader that CW microwave emission is generated with digital pulse train.

Thank you for the suggestion. We have added the description in the revised caption of this figure.

Revision: in the caption of Extended Data Fig. 4, we have added

...The CW output is prepared by using a train of δ function-like magnetic flux pulses to drive the signal source. ...

- Extended Data Fig.5: how is the flux calibrated? Is it just using the “dc-voltage to flux calibration” and applying it to the digital voltage setpoints? In this case, it is not surprising that pulse distortion plays a big role. There is probably a way to calibrate the actual flux waveform by monitoring the cavity frequency at the function of time.

We thank the referee for the nice comment. In principle, it is a measure of the instrument response and the device response to a fast flux. We only calibrated the instrument responses at room temperature for the coaxial cables, which were later used in the cryogenic experiments. We did not manage to measure the instrument response for the twisted pair of wires since their resistances are different at room temperature and cryogenic temperature.

The referee suggests using the time-varied cavity frequency as an oscilloscope to perform an in-situ calibration of the instrument response. We indeed tried this method, but it did not work well. This is mainly due to the bad signal-to-noise ratio of the cavity frequency measurement when integrating a very short period of the cavity signal. This short period is on the order of 10 nanoseconds given by the time resolution of the instrument response measurement.

- SI, after (S7): is the approximation valid considering the large flux change in this system?

We thank the referee for the insightful comment. Under the same parameter conditions as Fig. S2b, the maximum approximate angles during the photon generation process with different initial fluxes are shown in the figure below, while the end flux is fixed at 0. When the flux step changes crossing the boundary at half of the odd integer multiple of the magnetic flux quantum, the approximate angle $\frac{2\pi\Phi'}{\Phi_0} + \theta$ reaches its maximum. At this point, the approximate condition is not strictly satisfied.

In this work, we aim to use this toy model to illustrate the mechanism behind this emission rather than provide a strict numerical estimation. When the junction asymmetry δ is large, this approximate condition is nearly satisfied. In addition, we can predict the emission phenomenon with this toy model despite a smaller emission photon number when the flux step

changes crossing the flux boundary $0.5\phi_0$. Therefore, we believe that this toy model is effective and sufficient in terms of understanding the emission mechanism and finding directions for improvements.

Figure R2: Approximate angle dependence on the initial flux.

REVIEWERS' COMMENTS

Reviewer #1 (Remarks to the Author):

I thank the authors for their detailed discussion and replies. I believe the authors have addressed the comments from the reviewers satisfactorily.

Although the present work is far from an all-in-one solution to mitigate the scalability issue, the demonstrated pulse generator can be a promising building block in large scale superconducting quantum computing system.

I recommend the publication of the manuscript in Nature Communications.

Reviewer #2 (Remarks to the Author):

In the revised manuscript along with the response, the authors have provided appropriate answers to my questions. I hereby conclude that I recommend the publication of this work in Nature Communications. There are two remaining comments/suggestions:

1. Regarding the discussion of qubit driving, is the idea that driving and readout can be implemented with the same resonator/same set of resonators? In most experiments on superconducting qubits, the readout and the coupling are implemented using different resonators for several physical reasons. A brief discussion about how the number resonators per qubit would be useful.
2. The authors can consider extending their discussion about the comparison between the presented technique with other techniques, as has been discussed in their response to another referee.

Reviewer #3 (Remarks to the Author):

After reviewing the authors response to all the referee comments, I do think that this work is novel and interesting, but I'm not really convinced that it can solve the scaling problem. While this method seems to work just fine to drive the readout cavities, this is not really the scaling bottleneck. When it comes to the main bottle neck, driving a large number of qubits, the approach presented by authors is not really compelling:

- The basic calculation for Rabi rate using the existing device is only $<1\text{MHz}$ which is too slow, with no real option to increase that closer to the 100MHz required in transmon based systems.

- The alternative resonant method doesn't seem very scalable. Dynamically controlling the resonators frequency is known to be a tricky thing. It usually requires relatively high frequency lines ($\sim 1\text{GHz BW}$), and very careful waveform tuning to compensate for flux overshoot and so on. It would also add an extra line, which increase heat load and overall system complexity.

- The lack of control over the waveform (limited to decaying exponential) looks like a major hurdle when it comes to achieving high gate fidelity.

That said, I thought this work was in interesting take on a hard problem and could be interesting to a wide audience. The concepts presented here might be exploited in the future, and improvement could possibly make this technique viable. As such I still think this paper could warrant publication in Nature Communication. However, the paper would be a lot stronger with experimental evidence that this technique can drive qubits, including careful characterization of gate fidelities and spurious decoherence rates.

Reply to the Report of Referee 1

I thank the authors for their detailed discussion and replies. I believe the authors have addressed the comments from the reviewers satisfactorily.

Although the present work is far from an all-in-one solution to mitigate the scalability issue, the demonstrated pulse generator can be a promising building block in large scale superconducting quantum computing system.

I recommend the publication of the manuscript in Nature Communications.

We thank the referee for reviewing our work again and pointing out: “the demonstrated pulse generator can be a promising building block in large scale superconducting quantum computing system.” We greatly appreciate the reviewer’s very positive evaluation of our work and kind recommendation of our manuscript for publication in Nature Communications.

Reply to the Report of Referee 2

In the revised manuscript along with the response, the authors have provided appropriate answers to my questions. I hereby conclude that I recommend the publication of this work in Nature Communications. There are two remaining comments/suggestions:

We would like to thank the referee for reviewing our manuscript again and for the kind recommendation of our manuscript for publication in Nature Communications. We are extremely grateful for the careful review and all the valuable suggestions from the referee, which helped us improve our manuscript significantly.

1. Regarding the discussion of qubit driving, is the idea that driving and readout can be implemented with the same resonator/same set of resonators? In most experiments on superconducting qubits, the readout and the coupling are implemented using different resonators for several physical reasons. A brief discussion about how the number resonators per qubit would be useful.

Thanks for the nice comment. It is indeed possible to use the same resonator for qubit drive and readout, which would result in a more compact design than today's chip layout, as commented by the referee. It is even possible to realize qubit state initialization with the same resonator considering its tunable internal loss. We have added some more discussion on this aspect in the revised manuscript.

Revision: from line 321 to line 323, we have added

In addition, the same resonator can be used for qubit state readout and reset [44], which leads to a more compact design.

[44] M. D. Reed, B. R. Johnson, A. A. Houck, L. DiCarlo, J. M. Chow, D. I. Schuster, L. Frunzio, and R. J. Schoelkopf, Fast reset and suppressing spontaneous emission of a superconducting qubit, *Applied Physics Letters* 96, (2010).

2. The authors can consider extending their discussion about the comparison between the presented technique with other techniques, as has been discussed in their response to another referee.

We sincerely thank the referee for the suggestion. We believe that in the manuscript we have stated the (possible) advantages of the pulse generator, including phase coherence, simple control, small footprint, compatibility with superconducting quantum circuits, etc. We prefer not to directly compare this pulse generator with other technologies in the current manuscript, since we did not demonstrate a full spectrum of technologies used for superconducting quantum computers.

Reply to the Report of Referee 3

After reviewing the authors response to all the referee comments, I do think that this work is novel and interesting, but I'm not really convinced that it can solve the scaling problem. While this method seems to work just fine to drive the readout cavities, this is not really the scaling bottleneck. When it comes to the main bottle neck, driving a large number of qubits, the approach presented by authors is not really compelling:

We sincerely appreciate the referee for reviewing our manuscript again. We are extremely grateful for the careful review and the valuable suggestions from the referee. Those insightful comments provided us with a fresh perspective and improved our manuscript significantly.

- The basic calculation for Rabi rate using the existing device is only $<1\text{MHz}$ which is too slow, with no real option to increase that closer to the 100MHz required in transmon based systems.

Yes, we do agree with the referee that the emission power demonstrated here is still not strong enough for qubit drive with the conventional approach. Since the pulse generator is still at its initial stage, we believe there is plenty of room to optimize both the theory and the device design to increase the emission power.

- The alternative resonant method doesn't seem very scalable. Dynamically controlling the resonators frequency is known to be a tricky thing. It usually requires relatively high frequency lines ($\sim 1\text{GHz}$ BW), and very careful waveform tuning to compensate for flux overshoot and so on. It would also add an extra line, which increase heat load and overall system complexity.

We agree with the referee that qubit driving with a dynamically tuned resonator is quite a challenge to realize a high-fidelity gate. It is true that in the ideal case, compensating the instrument response to remove the flux distortion is preferred, which requires arbitrary waveform generation and cables with large bandwidth, as commented by the referee. In some ongoing works, we are testing some small bandwidth cable connections with some relatively stable flux response after the flux step to realize the proposed qubit drive scheme.

- The lack of control over the waveform (limited to decaying exponential) looks like a major hurdle when it comes to achieving high gate fidelity.

Yes, we do agree with the referee that for superconducting qubits with small anharmonicity, e.g., transmon, the limited waveform is a critical bottleneck to have a high gate fidelity. We still do not have a feasible solution to this possible problem. Maybe we have to turn to qubits with large anharmonicity.

That said, I thought this work was an interesting take on a hard problem and could be interesting to a wide audience. The concepts presented here might be exploited in the future, and improvement could possibly make this technique viable. As such I still think this paper could warrant publication in Nature Communication. However, the paper would be a lot stronger with experimental evidence that this technique can drive qubits, including careful characterization of gate fidelities and spurious decoherence rates.

We would like to thank the referee again for all the extremely helpful and insightful comments, which helped us a lot improve our manuscript. We are grateful to the referee for the kind recommendation of our manuscript for publication in Nature Communications